# Improving the Estimation of Instance-dependent Transition Matrix by using Self-supervised Learning

## Abstract

The *transition matrix* reveals the transition relationship between clean labels and noisy labels. It plays an important role in building statistically consistent classifiers. In real-world applications, the transition matrix is usually unknown and has to be estimated. It is a challenging task to accurately estimate the transition matrix, especially when it depends on the instance. Given that both instances and noisy labels are available, the major difficulty of learning the transition matrix comes from the missing of clean information. A lot of methods have been proposed to infer clean information. The self-supervised learning has demonstrated great success. These methods could even achieve comparable performance with supervised learning on some datasets but without requiring any labels during the training. It implies that these methods can efficiently infer clean labels. Motivated by this, in this paper, we have proposed a practical method that leverages self-supervised learning to obtain nearly clean labels to help the learning of instance-dependent transition matrix. Empirically, the proposed method has achieved state-of-the-art performance on different datasets.

## 1 Introduction

Recently, more researchers in the deep learning community place emphasis on learning with noisy labels (Jiang et al., 2018; Liu, 2021; Yao et al., 2021b; Bai et al., 2021; Ciortan et al., 2021). This is because manually annotating large-scale datasets is labor-intensive and time-consuming. Then some cheap but imperfect methods, e.g., crowdsourcing and web crawling, have been used to collect large-scale datasets which usually contain label errors. Existing work shows that training deep learning models on these datasets can lead to performance degeneration, because deep models can memorize the noisy labels easily (Han et al., 2018; Bai et al., 2021). How to improve the robustness of deep models when training data containing label errors becomes an important research topic.

To learn a classifier robust to label noise, there are two streams of methods, i.e., *statistically inconsistent* methods and *statistically consistent* methods. The statistically inconsistent methods mainly focus on designing heuristics to reduce the negative effect of label noise (Nguyen et al., 2019; Li et al., 2019; 2020; Wei et al., 2020; Bai et al., 2021; Yao et al., 2021a). These methods have demonstrated strong empirical performance but usually require expensive hyper-parameter tuning and do not provide statistical guarantees. To address the limitation, another stream of methods focusing on designing *classifier-consistent* algorithms (Liu & Tao, 2015; Patrini et al., 2017; Xia et al., 2020; Li et al., 2021) by exploiting the noise transition matrix $\boldsymbol{T}(x) \in \mathbb{R}^{C \times C}$, where $\boldsymbol{T}_{ij}(x) = P(\tilde{Y} = j | Y = i, X = x)$, $X$ denotes the random variable for instances or features, $\tilde{Y}$ denotes the noisy label, $Y$ denotes the clean label, and $C$ denotes the number of classes. When the transition matrix is given, the optimal classifier defined on the clean domain can be learned by utilizing noisy data only (Liu & Tao, 2015; Xia et al., 2019).

In real-world applications, the instance-dependent transition matrix $\boldsymbol{T}(x)$ is usually unknown and has to be learned. It is still a challenging task to accurately learn $\boldsymbol{T}(x)$ (Li et al., 2019; Yao et al., 2020). The reason is that to accurately learn $\boldsymbol{T}(x)$, the instance $X$, the noisy label $\tilde{Y}$ and the clean label $Y$ generally have to be given. However, for the dataset containing label errors, clean labels usually are not available. In general, without any other assumptions, to learn the transition matrix

for an instance, its clean-label information has to be given. Then existing methods hope some clean-label information can be inferred to learn $T(x)$ (Xia et al., 2019; Yang et al., 2022; Li et al., 2021). We will discuss the details in Section 2.

Recently, the classification model based on self-supervised learning has demonstrated comparable performance with supervised learning on some benchmark datasets (He et al., 2020; Niu et al., 2021). This implies that self-supervised learning has a strong ability to infer clean labels. Motivated by this, in this paper, we propose CoNL (Contrastive label-Noise Learning), which leverages the self-supervised technique to learn the instance-dependent transition matrix. In CoNL, it contains two main stages: contrastive co-selecting and constraint $T(x)$ revision which are as follows:

- We propose contrastive co-selecting, which utilizes the visual representations learned by contrastive learning to select confident examples without employing noisy labels. In such a way, the learned visual representations will be less influenced by label errors. To select confident examples defined on the clean domain, we learn two classifiers estimating $P(Y|X)$ and two transition matrices simultaneously by employing noisy labels and learned representations. We also encourage the two classifiers to have different learning abilities by training them with the representations obtained from strong and weak data augmentations, respectively. Then they can learn different types of confident examples and be robust to different noise rates. Combining two classifiers can obtain more confident examples.
- We propose constraint $T(x)$ revision, which refines the learned transition matrix by employing the selected confident examples. Based on the philosophy that the favorable transition matrix would make the classification risks on both clean data and noisy data small. We fine-tune $T(x)$ by encourage the loss w.r.t. the estimated $P(Y|X)$ and the estimated $P(\tilde{Y}|X)$ be small on the selected confident examples.

The empirical results for both transition-matrix learning and classification have demonstrated the strong performance with different types and levels of label noise on three synthetic IDN datasets (MNIST, CIFAR10, SVHN) and one real-world noisy dataset (CIFAR-10N). The rest of this paper is organized as follows. In Sec. 2, we review related work on label-noise learning especially modeling noisy labels and contrastive learning. In Sec. 3, we discuss how to leverage contrastive learning to learn the instant-dependent transition matrix better. In Sec. 4, we provide the empirical evaluations of the proposed method. In Sec. 5, we conclude our paper.

## 2  LABEL-NOISE LEARNING AND CONTRASTIVE LEARNING

**Problem setting.**  Let $\tilde{D}$ be the distribution of a noisy example $(X, \tilde{Y}) \in \mathcal{X} \times \{1, \ldots, C\}$, where $X$ denotes the variable of instances, $\tilde{Y}$ the variable of noisy labels, $\mathcal{X}$ the feature space, $\{1, \ldots, C\}$ the label space, and $C$ the size of classes. In learning with noisy labels, clean labels are not available, given a noisy training sample $\tilde{S} = \{x_i, \tilde{y}_i\}_{i=1}^N$ independently drawn from $\tilde{D}$, the aim is to learn a robust classifier from the sample $\tilde{S}$.

**The noise transition matrix $T(x)$.**  The transition matrix $T(x)$ has been widely used to model label-noise generation. The $ij$-th entry of the transition matrix $T_{ij}(x) = P(\tilde{Y} = j|Y = i, X = x)$ represents the possibility of the clean label $Y = i$ of instance $x$ flip to the noisy label $\tilde{Y} = j$. Existed methods can learn statistically consistent classifiers when the transition matrix is given (Liu & Tao, 2015; Goldberger & Ben-Reuven, 2017; Yu et al., 2018; Xia et al., 2019; 2020; Li et al., 2021). The reason is that, the clean-class posterior $P(Y|X)$ can be inferred by using the transition matrix and the noisy-class posterior $P(\tilde{Y}|X)$ (Patrini et al., 2017), i.e.,

$$T(x)[P(Y = 1|x), \ldots, P(Y = C|x)]^\top = [P(\tilde{Y} = 1|x), \ldots, P(\tilde{Y} = C|x)]^\top.$$

In general, the transition matrices are not given and need to be estimated. Without any other assumptions, to learn the transition matrix for an instance, its clean-label information has to be given (Xia et al., 2019; Yang et al., 2022)

**Learning the transition matrix $T(x)$.**  The clean-label information is crucial for learning the transition matrix. To learn transition matrices for all instances, 1). existing methods first learn some of the transition matrices in a training sample by inferring the clean-label information. 2). Then, by

making additional assumptions, the learned transition matrices can be used to help learn the other instance-dependent transition matrices.

Specifically, to learn some of the transition matrices in a training sample, existing methods try to infer clean-label information of some instances. Then the transition matrices of these instances can be learned. For example, if some instances can be identified which belong to a specific class almost surely (i.e., anchor points), the transition matrices of these instances can be learned (Liu & Tao, 2015). If *Bayes optimal labels* of some instances can be identified, their Bayes-label transition matrices can be learned (Yang et al., 2022). If clean-class posteriors are far from uniform (i.e., sufficiently scattered), the transition matrices *enclosing* $P(\tilde{Y}|X)$ with minimum volume is unique and can be learned (Li et al., 2021).

Once some of the transition matrices are learned, different assumptions have been proposed to utilize the learned transition matrices to help learn transition matrices of other instances. For example, the manifold assumption, where the instances that are close in manifold distance have similar transition matrices Cheng et al. (2022); class-dependent assumption, where instances with the same clean labels have the same transition matrices (Liu & Tao, 2015; Patrini et al., 2017; Li et al., 2021); part-dependent assumption (Xia et al., 2020), where the instances with similar parts have similar transition matrices.

**Contrastive learning.** Contrastive learning (Sermanet et al., 2018; Dwibedi et al., 2019; Chen et al., 2020a; He et al., 2020), which could learn *semantically meaningful* features without human annotations (Hadsell et al., 2006; Wu et al., 2018), is an important branch of unsupervised representative learning using methods related to the contrastive loss (Hadsell et al., 2006).

Existing methods have shown that semantically meaningful features are a very important characteristic in the human visual system. Humans usually use their existing knowledge of visual categories to learn about new categories of objects, where the visual categories are often encoded as high-level semantic attributes (Rosch et al., 1976; Su & Jurie, 2012). Contrastive learning, which helps in learning semantically meaningful features, is therefore very useful in inferring clean labels. Empirically, contrastive learning shows superior performance to other unsupervised-learning techniques on different datasets on the classification task (Chen et al., 2020b; Niu et al., 2021).

In this paper, we adopt an unsupervised instance discrimination-based representative learning approach, MoCo (He et al., 2020). The basic idea of contrastive learning is that the query representation should be similar to its matching key representation and dissimilar to other key representations, i.e., contrastive learning can be formulated as a dictionary look-up problem. Given an image $x$, the corresponding images obtained by using different augmentations are $x_q$ and $x_k$. The query representation generated by the backbone $f_\theta$ is $f_\theta(x_q)$, The corresponding key representation generated by another backbone $f_{\theta_{mo}}$ is $f_{\theta_{mo}}(x_k)$. The key representation will be stored in a queue. To learn the representation, for each iteration, MoCo optimize $\theta$ according to the following loss function (He et al., 2020):

$$\mathcal{L}_{mo} = -\frac{1}{N}\sum_{x_q}\log\frac{exp(f_\theta(x_q)\cdot f_{\theta_{mo}}(x_k)/\tau)}{exp(f_\theta(x_q)\cdot f_{\theta_{mo}}(x_k)/\tau) + \sum_{x_k'}exp(f_\theta(x_q)\cdot f_{\theta_{mo}}(x_k')/\tau)} \tag{1}$$

where $x_k'$ is another image different from $x$, $f_{\theta_{mo}}(x_k')$ is the key representation for $x_k'$, and $\tau$ is the temperature. Then $\theta_{mo}$ is updated by according to the parameter $\theta$, i.e., $\theta_{mo} \leftarrow \mu\theta_{mo} + (1-\mu)\theta$, where $\mu \in [0,1)$ is the hyper-parameter momentum.

## 3  CONTRASTIVE LABEL-NOISE LEARNING

Motivated by the success of contrastive learning on the classification task. Previous work shows that some methods based on contrastive learning even could achieve comparable performance with supervised methods on some datasets (He et al., 2020; Chen et al., 2020b). In this section, we introduce Contrastive label-Noise Learning (CoNL), which aims to effectively leverage the advantage of contrastive learning. An overview of the method is shown in Fig. 1 and described in Algo. 1.

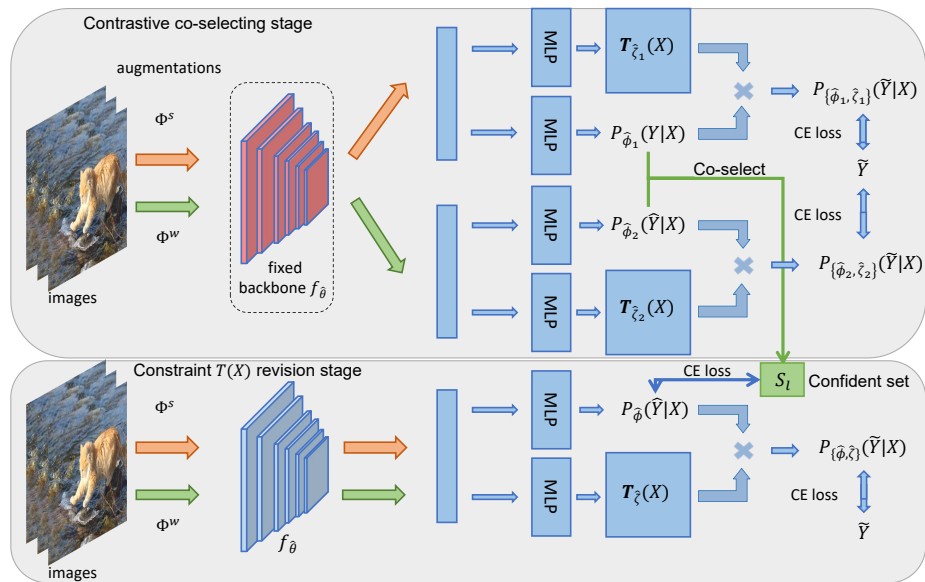

Figure 1: A working flow of our method CoNL.

## 3.1 LEVERAGING CONTRASTIVE LEARNING FOR CO-SELECTING

We aim to accurately select confident examples by leveraging contrastive learning. To achieve it, we utilize the contrastive method MOCO to learn the visual representations on training instances. Then, the learned representations obtained by applying strong and weak data augmentations are employed to learn two classifiers and transition metrics, respectively. To select confident examples, we exploit both the estimated clean-class posterior and consistency predicted clean labels of its neighbors. The details of contrastive co-selecting are as follows.

Firstly, to produce the visual representations, a backbone neural network $f_{\hat{\theta}} : \mathcal{X} \to \mathcal{Z}$ is trained by only using the training instance via Eq. (1). By employing the $f_{\hat{\theta}}$, we can obtain the representations $Z^s = \{z_i^s\}_{i=1}^n$ and $Z^w = \{z_i^w\}_{i=1}^n$ based on strong and weak data augmentations $\Phi^s$ and $\Phi^w$, respectively, where $z_i^s = f_{\hat{\theta}}(\Phi^s(x_i))$ and $z_i^w = f_{\hat{\theta}}(\Phi^w(x_i))$.

Let $g_{\phi_1}$ and $g_{\phi_2}$ be two classifier heads modeling $P_{\phi_1}(Y|X)$ and $P_{\phi_2}(Y|X)$ with learnable parameters $\phi_1$ and $\phi_2$, respectively. Let $\boldsymbol{T}_{\zeta_1}$ and $\boldsymbol{T}_{\zeta_2}$ be two transition matrices that modeled by neural networks with learnable parameters $\zeta_1$ and $\zeta_2$, respectively. To help learn transition matrix by employing the visual representations $Z^s$ and $Z^w$. We train two classifier heads $g_{\phi_1}$ and $g_{\phi_2}$ and two transition matrices $\boldsymbol{T}_{\zeta_1}$ and $\boldsymbol{T}_{\zeta_2}$ simultaneously on $Z_1$ and $Z_2$ by minimizing the cross-entropy loss, respectively. Specifically, the objective is as follows.

$$\{\hat{\phi}_1, \hat{\phi}_2, \hat{\zeta}_1, \hat{\zeta}_2\} = \underset{\phi_1, \phi_2, \zeta_1, \zeta_2}{\arg\min} \left( -\frac{1}{N} \sum_{i=1}^{N} \left( \tilde{y}_i \log(g_{\phi_1}(z_i^s) \boldsymbol{T}_{\zeta_1}(z_i^s)) + \tilde{y}_i \log(g_{\phi_2}(z_i^w) \boldsymbol{T}_{\zeta_2}(z_i^w)) \right) \right). \quad (2)$$

In this training process, the parameter of backbone $f_{\hat{\theta}}$ is fixed. There are several advantages. 1). By employing the representations that are independent of label errors, the classifiers can be better learned. Intuitively, previous work shows that representations learned by self-supervised learning contain the semantic information which usually correlates with clean labels (Wu et al., 2018; Niu et al., 2021). It implies that the representations contain some information about clean labels. In the training process, the visual representations are used as inputs of the classifier head. Then the learned classifier head also contains some information about these representations and clean labels. 2). By keeping $f_{\hat{\theta}}$ fixed, the learning difficulty of the transition matrix is reduced. In the learning process, only two simple models, i.e., the classifier head and the transition matrix, need to be learned. Since the visual representations could contain some information about clean labels, by also employing the noisy labels, the transition matrix can be effectively estimated.

Moreover, we train two classifier heads $g_{\phi_1}$ and $g_{\phi_2}$ with different data augmentations. In such a way, the two classifier heads are encouraged to be diverse and have different learning abilities. Then they can filter out different types of label errors and select different confident examples. We will

---

**Algorithm 1:** Contrastive label-Noise Learning (CoNL)

---

**Input:** A noisy training sample $\tilde{S}$, a noisy validation sample $\tilde{S}_v$

1 Get $f_{\hat{\theta}}$ by employing MoCo and all training instances in $\tilde{S}$ ;
2 Minimize the Eq. (2) to learn the parameters $\hat{\phi}_1, \hat{\phi}_2, \hat{\zeta}_1, \hat{\zeta}_2$ ;
3 Get the confident sample $S_l = S_l^w \bigcup S_l^s$, where $S_l^s$ and $S_l^w$ are generated according to Eq. (4) and Eq. (5) ;
4 Select the best transition matrix $\boldsymbol{T}_{\hat{\zeta}}$ and corresponding classifier head $f_{\hat{\theta}}$ by employing the validation sample $\tilde{S}_v$ ;
5 Get revised parameters $\hat{\theta}', \hat{\phi}'$ and $\hat{\zeta}'$ on the training sample $\tilde{S}$ and the confident sample $S_l$ by employing Eq. (7) ;
**Output:** $f_{\hat{\theta}'}, g_{\hat{\phi}'}, T_{\hat{\zeta}'}$

---

illustrate our confident sample-selection method on $g_{\hat{\phi}_1}$ which is trained on the representations with strong augmentations. The same sample-selection method are employed for $g_{\hat{\phi}_2}$.

Specifically, the trained classifier head $g_{\hat{\phi}_1}$ is employed to relabel all instances and get a set of examples $S^s = \{(x_i, \tilde{y}_i, \hat{y}_i^s)\}_{i=1}^N$. To select confident examples, we employ the combination of two criteria. The basic idea is that an instance is reliable if 1). The confidence of the predicted clean label is high 2). Its corresponding predicted clean labels are consistent with its neighbor examples' predicted clean labels.

To determine whether an example $(x_i, \tilde{y}_i, \hat{y}_i^s)$ should be the confident example or not, the confidence of the predicted clean label and the predicted clean label consistency of $x_i$ have to be calculated. The confidence of the predicted clean labels can be directly obtained via $g_{\hat{\phi}_1}(x_i)_{\hat{y}_i^s}$ which is the $\hat{y}_i^s$-th coordinate of the output $g_{\hat{\phi}_1}(x_i)$. The predicted clean label consistency $r_i$ of $x_i$ is calculated as follows.

$$r_i^s = \frac{1}{K^s} \sum_{y \in \mathcal{N}_i^s} \mathbb{1}(y = \hat{y}_i), \tag{3}$$

where $\mathcal{N}^s(x_i)$ contains neighbors of the instance $x_i$, when the strong augmentations are applied. It is obtained according to the cosine similarity of features extracted from the backbone. The $K^s$ nearest neighbors with high cosine similarity are selected as the neighbors of $x_i$.

By combining two criteria together, the example $(x_i, \tilde{y}_i, \hat{y}_i)$ is considered to be an confident example if both criteria $g_{\hat{\phi}_1}(x_i)_{\hat{y}_i^s} > \lambda$ and $r_i > \tau$ satisfy, where $\lambda$ and $\tau$ are hyper-parameters. Finally, the confident sample $S_l^s$ selected by the trained classifier heads $g_{\hat{\phi}_1}$ is as follows.

$$S_l^s = \{(x_i, \tilde{y}_i, \hat{y}_i) | r_i > \tau, g_{\hat{\phi}_1}(x_i)_{\hat{y}_i^s} > \lambda, \forall i = 1, 2, \ldots, N\}. \tag{4}$$

By applying the weak data augmentation on training instances and using the same selection method for $g_{\hat{\phi}_2}$, the confident sample $S_l^w$ selected by $g_{\hat{\phi}_2}$ can also be obtained, i.e.,

$$S_l^w = \{(x_i, \tilde{y}_i, \hat{y}_i) | r_i > \tau, g_{\hat{\phi}_1}(x_i)_{\hat{y}_i^w} > \lambda, \forall i = 1, 2, \ldots, N\}. \tag{5}$$

Then, to utilize different confident examples obtained by applying different data augmentations, we union two confident samples, i.e., $S_l = S_l^w \bigcup S_l^s$ which will be used for constraint $\boldsymbol{T}(x)$ revision.

## 3.2 CONSTRAINT $\boldsymbol{T}(x)$ REVISION

We improve $\boldsymbol{T}$-revision (Xia et al., 2019) by utilizing the confident sample $S_l$ to refine the transition matrix which depends on the instance. The philosophy of constraint $\boldsymbol{T}(x)$ revision is that the favorable transition matrix would make the classification losses on both clean labels and noisy labels small. Once we have the confident sample $S_l$ containing both noisy labels and predicted clean labels. We could regularize and refine the transition matrix by minimizing classification losses of both noisy labels and predicted clean labels of the selected confident examples. After fine-tuning the transition matrix, it can also help learn better the representation and the classifier head. Therefore, in this stage, we also fine-tune the representation and the classifier head.

Specifically, By comparing the the validation accuracy of $g_{\hat{\phi}_1}(\cdot)\boldsymbol{T}_{\hat{\zeta}_1}(\cdot)$ and $g_{\hat{\phi}_2}(\cdot)\boldsymbol{T}_{\hat{\zeta}_2}(\cdot)$, we select the best transition matrix and the classifier which denote as $T_{\hat{\zeta}}$ and $g_{\hat{\phi}}$, respectively. Let $M$ be the number of confident examples. Reminding that $\Phi^s$ and $\Phi^w$ are strong and weak data augmentations, respectively. To minimize the classification loss $\mathcal{L}_c$ on the predicted clean labels, the confident sample $S_l$ is employed. The loss function is as follows.

$$\mathcal{L}_{\hat{Y}}(S_l, g_{\hat{\phi}}, f_{\hat{\theta}}) = -\frac{1}{M} \sum_{(x,\tilde{y},\hat{y}) \in S_l} \left( \hat{y} \log(g_{\hat{\phi}}(f_{\hat{\theta}}(\Phi^s(x)))) + \hat{y} \log(g_{\hat{\phi}}(f_{\hat{\theta}}(\Phi^w(x)))) \right). \quad (6)$$

To minimize the classification loss on noisy labels, the noisy training sample $\tilde{S}$ is employed, i.e.,

$$\mathcal{L}_{\tilde{Y}}(\tilde{S}, g_{\hat{\phi}}, f_{\hat{\theta}}, \boldsymbol{T}_{\hat{\zeta}}) = -\frac{1}{N} \sum_{(x,\tilde{y}) \in \tilde{S}} \left( \tilde{y} \log(g_{\hat{\phi}}(f_{\hat{\theta}}(\Phi^s(x)))\boldsymbol{T}_{\hat{\zeta}}\left(f_{\hat{\theta}}(\Phi^s(x))\right)) \right.$$
$$\left. + \tilde{y} \log(g_{\hat{\phi}}(f_{\hat{\theta}}(\Phi^w(x)))\boldsymbol{T}_{\hat{\zeta}}(f_{\hat{\theta}}(\Phi^w(x)))) \right).$$

By combining two losses $\mathcal{L}_{\hat{Y}}$ and $\mathcal{L}_{\tilde{Y}}$ together, we fine-tune the transition matrix $T_{\hat{\zeta}}$, the classification head $g_{\hat{\phi}}$ and the backbone model $f_{\hat{\theta}}$ by employing the objective function:

$$\{\hat{\theta}', \hat{\phi}', \hat{\zeta}'\} = \underset{\hat{\theta}, \hat{\phi}, \hat{\zeta}}{\arg\min} \left( \mathcal{L}_{\hat{Y}}(S_l, g_{\hat{\phi}}, f_{\hat{\theta}}) + \mathcal{L}_{\tilde{Y}}(\tilde{S}, g_{\hat{\phi}}, f_{\hat{\theta}}, T_{\hat{\zeta}}) \right). \quad (7)$$

In Section 4, we show that by employing constraint $\boldsymbol{T}(X)$ revision, both the estimation of the transition matrix and the classification accuracy can be dramatically improved.

## 4 EXPERIMENTS

In this section, we present the empirical results of the proposed method and *state-of-the-art* methods on synthetic and real-world noisy datasets. We also conduct ablation studies to demonstrate the effectiveness of contrastive co-selecting and constraint $\boldsymbol{T}(x)$ revision.

### 4.1 EXPERIMENT SETUP

**Datasets.** We empirically verify the performance of our method on three synthesis datasets, i.e., Fashion-MNIST (Xiao et al., 2017), SVHN (Netzer et al., 2011), CIFAR-10 (Krizhevsky et al., 2009), and one real-world dataset, i.e., CIFAR-10N (Wei et al., 2022). Fashion-MNIST contains 70,000 28x28 grayscale images with 10 classes total, and 60,000 images for training and 10,000 images for testing. SVHN contains 73,257 training images and 26,032 testing images. CIFAR-10 contains 50,000 training images and 10,000 testing images. Both SVHN and CIFAR-10 have 10 classes of images, and the image size is 32x32. The three datasets contain clean labels. We corrupted the training data manually according to the instance-dependent label noise generation method proposed in (Xia et al., 2020). All experiments are repeated five times. CIFAR-10N is a real-world label-noise version of CIFAR-10, it contains human-annotated noisy labels with five different types of noise (Worst, Aggregate, Random 1, Random 1, and Random 3). For all datasets, we leave out 10% of training examples as a noisy validation set.

**Baselines.** The baselines used in our experiments for comparison are: 1). CE, training the model using standard cross-entropy loss on noisy data directly; 2), GCE (Zhang & Sabuncu, 2018), which use standard cross-entropy loss and mean absolute error to train model on noisy data; 3), Mentor-Net (Jiang et al., 2018), which pretrains a model to select reliable samples for the main model; 4), Co-teaching (Han et al., 2018), which trains two models simultaneously to select reliable samples for each other; 5), Reweight (Liu & Tao, 2015), which exploits importance reweighting method to estimate a unbiased risk defined on clean data using noisy data; 6), Forward (Patrini et al., 2017), using class-dependent transition matrix to correct loss function; 7), PTD (Xia et al., 2020), estimating instance-dependent transition matrix through part-dependent transition matrices; 8) CausalNL (Yao et al., 2021b), which explores using causal mechanism to excavate clean-label information on noisy data; 9), MEIDTM (Cheng et al., 2022), which uses Lipschitz continuity to constrain the transition matrix; 10), BLTM (Yang et al., 2022), which using Bayes optimal label to learn instance-dependent transition matrix; 11), NPC (Bae et al., 2022), which proposes a post-processing scheme to calibrate the prediction of a noise-robust classifier.

Table 1: Means and standard deviations (percentage) of classification accuracy on Fashion-MNIST.

|  | IDN-10% | IDN-20% | IDN-30% | IDN-40% | IDN-50% |
|---|---|---|---|---|---|
| CE | $93.16 \pm 0.02$ | $92.68 \pm 0.16$ | $91.41 \pm 0.16$ | $87.60 \pm 0.33$ | $71.76 \pm 0.77$ |
| GCE | $92.52 \pm 0.14$ | $92.52 \pm 0.09$ | $91.49 \pm 0.09$ | $88.49 \pm 0.23$ | $76.29 \pm 0.82$ |
| MentorNet | $93.16 \pm 0.01$ | $91.57 \pm 0.29$ | $90.52 \pm 0.41$ | $88.14 \pm 0.76$ | $61.62 \pm 1.42$ |
| CoTeaching | $94.26 \pm 0.06$ | $91.21 \pm 0.31$ | $90.30 \pm 0.42$ | $89.10 \pm 0.29$ | $63.22 \pm 1.56$ |
| Reweight | $93.42 \pm 0.16$ | $93.12 \pm 0.18$ | $92.19 \pm 0.18$ | $88.51 \pm 1.52$ | $75.00 \pm 5.28$ |
| Forward | $93.48 \pm 0.11$ | $92.82 \pm 0.12$ | $91.05 \pm 1.44$ | $87.82 \pm 1.81$ | $78.34 \pm 2.98$ |
| PTD | $91.55 \pm 2.47$ | $87.68 \pm 5.50$ | $82.78 \pm 5.29$ | $75.21 \pm 1.81$ | $66.21 \pm 5.15$ |
| CausalNL | $91.63 \pm 0.18$ | $90.84 \pm 0.31$ | $90.68 \pm 0.37$ | $90.01 \pm 0.45$ | $78.19 \pm 1.01$ |
| BLTM | $91.28 \pm 1.93$ | $91.20 \pm 0.27$ | $85.51 \pm 4.77$ | $82.42 \pm 1.51$ | $67.65 \pm 5.65$ |
| MEIDTM | $86.00 \pm 0.84$ | $80.99 \pm 0.64$ | $73.12 \pm 2.34$ | $63.81 \pm 3.02$ | $58.60 \pm 3.32$ |
| NPC | $88.78 \pm 0.30$ | $88.05 \pm 0.02$ | $84.99 \pm 1.20$ | $82.59 \pm 1.22$ | $70.58 \pm 4.43$ |
| CoNL-NR | $92.14 \pm 0.07$ | $91.86 \pm 0.26$ | $91.40 \pm 0.15$ | $90.09 \pm 0.41$ | $85.84 \pm 0.71$ |
| CoNL | $\mathbf{94.98 \pm 0.14}$ | $\mathbf{94.20 \pm 0.23}$ | $\mathbf{92.92 \pm 1.05}$ | $\mathbf{91.01 \pm 1.83}$ | $\mathbf{86.01 \pm 1.83}$ |

Table 2: Means and standard deviations (percentage) of classification accuracy on SVHN.

|  | IDN-10% | IDN-20% | IDN-30% | IDN-40% | IDN-50% |
|---|---|---|---|---|---|
| CE | $95.75 \pm 0.07$ | $95.33 \pm 0.18$ | $93.74 \pm 0.29$ | $91.13 \pm 0.32$ | $72.43 \pm 2.78$ |
| GCE | $95.09 \pm 0.03$ | $94.37 \pm 0.13$ | $86.39 \pm 5.12$ | $81.95 \pm 1.45$ | $63.20 \pm 2.75$ |
| MentorNet | $95.54 \pm 0.12$ | $94.76 \pm 0.16$ | $92.39 \pm 0.18$ | $90.41 \pm 0.49$ | $61.23 \pm 2.82$ |
| CoTeaching | $94.66 \pm 0.36$ | $93.93 \pm 0.31$ | $92.06 \pm 0.31$ | $91.93 \pm 0.81$ | $67.62 \pm 1.99$ |
| Reweight | $95.91 \pm 0.44$ | $94.23 \pm 2.53$ | $91.06 \pm 4.09$ | $87.92 \pm 6.46$ | $85.30 \pm 0.10$ |
| Forward | $96.12 \pm 0.11$ | $95.84 \pm 0.07$ | $94.07 \pm 2.14$ | $87.38 \pm 3.85$ | $82.02 \pm 4.81$ |
| PTD | $72.90 \pm 1.31$ | $75.68 \pm 9.43$ | $75.01 \pm 1.98$ | $31.59 \pm 5.58$ | $30.58 \pm 2.32$ |
| CausalNL | $94.20 \pm 0.09$ | $94.06 \pm 0.23$ | $93.86 \pm 0.37$ | $\mathbf{93.82 \pm 0.45}$ | $85.41 \pm 2.95$ |
| BLTM | $93.88 \pm 0.55$ | $92.66 \pm 1.53$ | $92.18 \pm 0.61$ | $84.33 \pm 5.44$ | $76.19 \pm 5.17$ |
| MEIDTM | $94.76 \pm 0.10$ | $93.56 \pm 0.22$ | $91.11 \pm 0.42$ | $86.11 \pm 0.34$ | $72.66 \pm 2.50$ |
| NPC | $94.36 \pm 0.06$ | $93.41 \pm 0.07$ | $90.31 \pm 0.59$ | $84.81 \pm 1.80$ | $70.15 \pm 0.76$ |
| CoNL-NR | $92.64 \pm 0.17$ | $92.84 \pm 0.03$ | $92.48 \pm 0.16$ | $89.68 \pm 1.35$ | $78.90 \pm 3.05$ |
| CoNL | $\mathbf{96.81 \pm 0.09}$ | $\mathbf{96.29 \pm 0.47}$ | $\mathbf{95.37 \pm 0.88}$ | $91.73 \pm 2.53$ | $\mathbf{86.51 \pm 1.14}$ |

**Implementation.** We implement our algorithm using PyTorch and conduct all our experiments on RTX 3090. We use a ResNet-18 as the backbone for Fashion-MNIST, a ResNet-34 as the backbone for SVHN and CIFAR-10. For the classifier head and transition matrix generator, we use a two layers MLP with ReLU activation function. The final layer of the transition matrix generator is initialized to generate a diagonal largest transition matrix. To learn the backbone, we follow the same settings of MoCo (He et al., 2020). We set the temperature $\tau = 0.2$ and set $\mu = 0.999$. The size of the queue is 12800. Total epochs are 1000. When training the classifier heads and transition matrices, we use SGD with momentum 0.9, weight decay $10^{-4}$, batch size 128 and an initial learning rate of $10^{-2}$ to optimize the networks. The learning rate is divided by 10 at the 5th epochs and 7th epochs. We set 10 epochs in total. When revising the transition matrix, we use Adam with an initial learning rate $10^{-4}$. The learning rate is divided by 10 at the 5th epochs and 7th epochs. We set 10 epochs in total. After that, we optimize the backbone and classifier head using SGD with momentum 0.9, weight decay $10^{-4}$, batch size 128 and an initial learning rate of $10^{-2}$. At the same time, we optimize the transition matrix generator using Adam with an initial learning rate $10^{-4}$. The learning rate is divided by 10 at the 10th epochs and 20th epochs. We set 30 epochs in total.

### 4.2 CLASSIFICATIONS ACTUARIES ON DIFFERENT DATASETS

We conduct experiments on Fashion-MNIST, SVHN, CIFAR-10 and CIFAR-10N. The noise rate for Fashion-MNIST, SVHN and CIFAR-10 is ranged from 0.1 to 0.5. The ones with CoNL-NR represent the results of our algorithm without constraint $\boldsymbol{T}(X)$ revision. The ones with CoNL represent the results after constraint $\boldsymbol{T}(X)$ revision. As shown in Tab. 1, Tab. 2 and Tab. 3, The proposed method is have outperform to other methods by a large margin when the noise rate is large.

We also conduct experiments on the real-world label-noise dataset CIFAR-10N. The results of full types of noise (Worst, Aggregate, Random 1, Random 1, and Random 3) are shown in Tab. 4. The experiment results show that our method also works well on the real-world label-noise dataset.

Table 3: Means and standard deviations (percentage) of classification accuracy on CIFAR-10.

|         | IDN-10%         | IDN-20%         | IDN-30%           | IDN-40%           | IDN-50%          |
|---------|-----------------|-----------------|-------------------|-------------------|------------------|
| CE        | 87.81 ± 0.15  | 85.90 ± 0.30  | 82.67 ± 0.31    | 74.49 ± 0.95    | 46.81 ± 2.52   |
| GCE       | 84.24 ± 0.38  | 80.22 ± 1.30  | 69.31 ± 0.18    | 56.86 ± 0.92    | 53.44 ± 1.28   |
| MentorNet | 86.87 ± 0.14  | 83.89 ± 0.16  | 77.83 ± 0.28    | 61.96 ± 0.97    | 47.89±2.03     |
| CoTeaching| 90.06 ± 0.32  | 87.16 ± 0.50  | 81.80 ± 0.26    | 63.95 ± 2.87    | 45.92±2.21     |
| Reweight  | 89.63 ± 0.27  | 87.85 ± 0.97  | 81.29 ± 6.49    | 80.33 ± 3.75    | 75.14 ± 2.40   |
| Forward   | 88.89 ± 0.18  | 87.83 ± 0.30  | 82.01 ± 3.29    | 79.49 ± 1.85    | 71.11 ± 8.78   |
| PTD       | 73.50 ± 3.04  | 71.64 ± 3.13  | 64.34 ± 12.38   | 62.53 ± 10.93   | 51.04 ± 8.28   |
| CausalNL  | 83.39 ± 0.34  | 80.91 ± 1.14  | 79.05 ± 0.54    | 79.08 ± 0.50    | 76.56 ± 0.02   |
| BLTM      | 80.16 ± 0.37  | 77.50 ± 1.30  | 71.47 ± 2.33    | 63.20 ± 4.52    | 48.12 ± 9.03   |
| MEIDTM    | 86.52 ± 0.38  | 82.93 ± 0.44  | 77.35 ± 0.21    | 68.21 ± 2.09    | 57.84 ± 3.51   |
| NPC       | 84.83 ± 0.22  | 83.13 ± 0.28  | 79.48 ± 0.31    | 73.85 ± 0.41    | 67.04 ± 0.06   |
| CoNL-NR   | 90.04 ± 0.18  | 90.23 ± 0.16  | 89.44 ± 0.30    | 87.03 ± 1.41    | 74.14 ± 6.46   |
| CoNL      | **94.37 ± 0.16** | **93.88 ± 0.17** | **91.37 ± 3.38** | **89.09 ± 3.05** | **84.79 ± 1.63** |

Table 4: Means and standard deviations (percentage) of classification accuracy on CIFAR-10N.

|           | Worst          | Aggregate      | Random 1        | Random 2        | Random 3        |
|-----------|----------------|----------------|-----------------|-----------------|-----------------|
| CE        | 79.39 ± 0.35 | 87.91 ± 0.18 | 86.05 ± 0.13  | 86.12 ± 0.12  | 86.12 ± 0.16  |
| GCE       | 75.45 ± 0.31 | 82.77 ± 0.13 | 81.18 ± 0.22  | 80.39 ± 0.54  | 80.89 ± 0.82  |
| MentorNet | 77.91 ± 0.38 | 75.56 ± 0.25 | 77.12 ± 0.05  | 76.03 ± 0.81  | 76.57 ± 0.18  |
| CoTeaching| 81.86 ± 0.40 | 82.45 ± 0.08 | 82.90 ± 0.46  | 82.95 ± 0.26  | 82.66 ± 0.12  |
| Reweight  | 77.68 ± 2.46 | 89.34 ± 0.09 | 88.44 ± 0.001 | 88.15 ± 0.13  | 88.21 ± 0.28  |
| Forward   | 79.27 ± 1.18 | 89.22 ± 0.21 | 86.84 ± 0.97  | 86.99 ± 0.10  | 87.63 ± 0.39  |
| PTD       | 67.72 ± 3.33 | 78.20 ± 0.25 | 84.19 ± 0.51  | 84.36 ± 0.55  | 69.31 ± 0.43  |
| CausalNL  | 73.09 ± 2.01 | 82.74 ± 0.38 | 80.80 ± 0.50  | 81.26 ± 0.69  | 81.02 ± 1.48  |
| BLTM      | 68.21 ± 1.67 | 79.41 ± 1.00 | 78.09 ± 1.03  | 76.99 ± 1.23  | 76.34 ± 0.58  |
| MEIDTM    | 79.59 ± 0.89 | 90.15 ± 0.27 | 87.81 ± 0.52  | 88.07 ± 0.18  | 87.86 ± 0.21  |
| NPC       | 75.53 ± 1.64 | 84.44 ± 0.47 | 81.87 ± 0.80  | 81.47 ± 0.95  | 82.22 ± 0.58  |
| CoNL-NR   | 86.52 ± 0.19 | 89.81 ± 0.31 | 89.63 ± 0.19  | 89.43 ± 0.27  | 89.07 ± 0.33  |
| CoNL      | **87.07 ± 0.68** | **93.17 ± 0.17** | **91.65 ± 0.32** | **91.51 ± 0.15** | **91.28 ± 0.36** |

## 4.3 ABLATION STUDIES

We perform ablation studies on Fashion-MNIST, SVHN and CIFAR10 including the performance of contrastive co-selecting, constraint $T(x)$ revision and the influence of contrastive learning on the accuracy. We leave results on Fashion-MNIST in our appendix due to the limited space.

### 4.3.1 CONFIDENT EXAMPLE SELECTION

We illustrate the performance of contrastive co-selecting in Tab. 5, Tab. 6 and Tab. 11. The results demonstrate that contrastive co-selecting can accurately select confident examples. Specifically, under $50\%$ of instance-dependent noise, it can select at least $35.56\%$ of examples with the clean ratio of at least $97.88\%$. Moreover, the noise rate of the selected example set $S_l$ is also close to the real noise rate. This could make a great contribution to the revision of transition matrix $T(x)$.

### 4.3.2 THE ESTIMATION ERROR OF THE TRANSITION MATRIX

We show the estimation error of transition matrix before and after constraint $T(X)$ revision. To caculate the estimation error, we compare the difference between the ground-truth transition matrix and the estimated transition matrix by employing $l_1$ norm For each instance, we only analyze the estimation error of a specific low since the noise is generated by one row of $T(x)$. The experiment results are showed in Tab. 7, Tab. 8 and Tab. 13. The results show that constraint $T(X)$ revision can effectively reduce the estimation error of transition matrix.

Table 5: Performance of contrastive co-selecting on SVHN.

|  | IDN-10% | IDN-20% | IDN-30% | IDN-40% | IDN-50% |
|---|---|---|---|---|---|
| selection ratio | $37.18 \pm 0.49$ | $37.71 \pm 0.16$ | $37.25 \pm 0.09$ | $36.85 \pm 0.19$ | $35.56 \pm 0.57$ |
| Noise rate | $13.31 \pm 0.12$ | $21.01 \pm 0.07$ | $30.63 \pm 0.08$ | $40.29 \pm 0.14$ | $49.86 \pm 0.30$ |
| clean ratio | $99.40 \pm 0.02$ | $99.38 \pm 0.02$ | $99.40 \pm 0.03$ | $99.33 \pm 0.13$ | $97.88 \pm 2.82$ |

Table 6: Performance of contrastive co-selecting on CIFAR-10.

|  | IDN-10% | IDN-20% | IDN-30% | IDN-40% | IDN-50% |
|---|---|---|---|---|---|
| selection ratio | $66.33 \pm 2.33$ | $69.06 \pm 0.84$ | $63.54 \pm 0.47$ | $57.70 \pm 0.99$ | $47.90 \pm 0.83$ |
| Noise rate | $13.64 \pm 0.09$ | $21.37 \pm 0.06$ | $31.49 \pm 2.10$ | $40.53 \pm 0.22$ | $49.73 \pm 0.12$ |
| clean ratio | $99.14 \pm 0.04$ | $99.15 \pm 0.05$ | $96.89 \pm 4.83$ | $98.45 \pm 1.95$ | $99.18 \pm 0.36$ |

Table 7: Transition matrix estimation error on SVHN.

|  | IDN-10% | IDN-20% | IDN-30% | IDN-40% | IDN-50% |
|---|---|---|---|---|---|
| CoNL (w/o MoCo) | $0.193 \pm 0.008$ | $0.204 \pm 0.020$ | $0.229 \pm 0.026$ | $0.251 \pm 0.001$ | $0.285 \pm 0.014$ |
| CoNL-NR | $0.256 \pm 0.061$ | $0.193 \pm 0.006$ | $0.214 \pm 0.006$ | $0.240 \pm 0.008$ | $0.305 \pm 0.015$ |
| CoNL | $\mathbf{0.177 \pm 0.020}$ | $\mathbf{0.188 \pm 0.005}$ | $\mathbf{0.206 \pm 0.009}$ | $\mathbf{0.226 \pm 0.011}$ | $\mathbf{0.231 \pm 0.019}$ |

Table 8: Transition matrix estimation error on CIFAR-10.

|  | IDN-10% | IDN-20% | IDN-30% | IDN-40% | IDN-50% |
|---|---|---|---|---|---|
| CoNL (w/o MoCo) | $0.217 \pm 0.006$ | $0.221 \pm 0.008$ | $0.253 \pm 0.003$ | $0.303 \pm 0.001$ | $0.384 \pm 0.003$ |
| CoNL-NR | $0.206 \pm 0.011$ | $0.198 \pm 0.001$ | $0.252 \pm 0.029$ | $0.285 \pm 0.022$ | $0.350 \pm 0.018$ |
| CoNL | $\mathbf{0.168 \pm 0.009}$ | $\mathbf{0.195 \pm 0.003}$ | $\mathbf{0.229 \pm 0.019}$ | $\mathbf{0.255 \pm 0.009}$ | $\mathbf{0.262 \pm 0.005}$ |

Table 9: The test accuracy of with MoCo and without MoCo on SVHN.

|  | IDN-10% | IDN-20% | IDN-30% | IDN-40% | IDN-50% |
|---|---|---|---|---|---|
| CoNL (w/o MoCo) | $94.88 \pm 0.13$ | $94.56 \pm 0.12$ | $93.85 \pm 0.24$ | $86.47 \pm 1.17$ | $82.75 \pm 2.33$ |
| CoNL-NR | $92.64 \pm 0.17$ | $92.84 \pm 0.03$ | $92.48 \pm 0.16$ | $89.68 \pm 1.35$ | $78.90 \pm 3.05$ |
| CoNL | $\mathbf{96.81 \pm 0.09}$ | $\mathbf{96.29 \pm 0.47}$ | $\mathbf{95.37 \pm 0.88}$ | $\mathbf{91.73 \pm 2.53}$ | $\mathbf{86.51 \pm 1.14}$ |

Table 10: The test accuracy of with MoCo and without MoCo on CIFAR-10.

|  | IDN-10% | IDN-20% | IDN-30% | IDN-40% | IDN-50% |
|---|---|---|---|---|---|
| CoNL (w/o MoCo) | $83.88 \pm 0.17$ | $82.53 \pm 0.46$ | $80.51 \pm 0.22$ | $75.34 \pm 0.62$ | $68.62 \pm 3.56$ |
| CoNL-NR | $90.04 \pm 0.18$ | $90.23 \pm 0.16$ | $89.44 \pm 0.30$ | $87.03 \pm 1.41$ | $74.14 \pm 6.46$ |
| CoNL | $\mathbf{94.37 \pm 0.16}$ | $\mathbf{93.88 \pm 0.17}$ | $\mathbf{91.37 \pm 3.38}$ | $\mathbf{89.09 \pm 3.05}$ | $\mathbf{84.79 \pm 1.63}$ |

### 4.3.3 ACCURACIES WITH OR WITHOUT EMPLOYING CONTRASTIVE LEARNING

We conduct the experiments with or without employing contrastive learning on different datasets. For CoNL (w/o MoCo), the backbone method $f_\theta$ is not trained by employing MoCo, but during the co-selecting stage, we do not fix its parameter. The other settings are same as CoNL. The experiment results are showed in Tab. 9, Tab. 10 and Tab. 12. The empirical results clearly show that the contrastive learning technique dramatically improves the robustness of the learning model and is powerful for inferring the clean-label information.

## 5 CONCLUSION

Since both instances and noise labels are available, the main difficulty in learning instance-dependent transition matrices is due to the lack of clean-label information. Motivated by the great success of self-supervised learning in inferring clean labels. In this article, we propose CoNL (Contrastive Label-Noise Learning), which can effectively utilize self-supervised learning to learn instance-dependent transition matrices. Empirically, the proposed method have achieved state-of-the-art performance on different datasets.

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

# A ABLATION STUDIES ON FASHION-MNIST

We demonstrate results of ablation studies on Fashion-MNIST including the performance of contrastive co-selecting, constraint $T(x)$ revision and the influence of contrastive learning on the accuracy in Tab. 11, Tab. 13 and Tab. 12, respectively.

Table 11: Performance of contrastive co-selecting on Fashion-MNIST.

|  | IDN-10% | IDN-20% | IDN-30% | IDN-40% | IDN-50% |
|---|---|---|---|---|---|
| selection ratio | $23.28 \pm 0.84$ | $26.00 \pm 1.09$ | $23.46 \pm 1.10$ | $22.37 \pm 1.42$ | $11.94 \pm 1.87$ |
| Noise rate | $11.82 \pm 0.20$ | $20.18 \pm 0.13$ | $30.12 \pm 0.21$ | $39.47 \pm 0.26$ | $49.21 \pm 0.42$ |
| clean ratio | $99.76 \pm 0.11$ | $99.81 \pm 0.06$ | $99.88 \pm 0.03$ | $99.87 \pm 0.06$ | $99.74 \pm 0.32$ |

Table 12: Test accuracy of with MoCo and without MoCo on Fashion-MNIST.

|  | IDN-10% | IDN-20% | IDN-30% | IDN-40% | IDN-50% |
|---|---|---|---|---|---|
| CoNL (w/o MoCo) | $91.79 \pm 0.15$ | $90.76 \pm 0.14$ | $87.46 \pm 2.32$ | $84.43 \pm 1.43$ | $75.16 \pm 5.13$ |
| CoNL-NR | $92.14 \pm 0.07$ | $91.86 \pm 0.26$ | $91.40 \pm 0.15$ | $90.09 \pm 0.41$ | $85.84 \pm 0.71$ |
| CoNL | $\mathbf{94.98 \pm 0.14}$ | $\mathbf{94.20 \pm 0.23}$ | $\mathbf{92.92 \pm 1.05}$ | $\mathbf{91.01 \pm 1.83}$ | $\mathbf{86.01 \pm 1.83}$ |

Table 13: Transition matrix estimation error on Fashion-MNIST.

|  | IDN-10% | IDN-20% | IDN-30% | IDN-40% | IDN-50% |
|---|---|---|---|---|---|
| CoNL (w/o MoCo) | $0.247 \pm 0.005$ | $0.360 \pm 0.012$ | $0.450 \pm 0.011$ | $0.582 \pm 0.023$ | $0.674 \pm 0.020$ |
| CoNL-NR | $0.236 \pm 0.003$ | $0.327 \pm 0.021$ | $0.409 \pm 0.010$ | $0.528 \pm 0.013$ | $0.674 \pm 0.018$ |
| CoNL | $0.238 \pm 0.006$ | $\mathbf{0.324 \pm 0.005}$ | $\mathbf{0.391 \pm 0.008}$ | $\mathbf{0.467 \pm 0.013}$ | $\mathbf{0.540 \pm 0.003}$ |

# B EXPERIMENTS ON CIFAR-100 AND WEBVISION

To verify whether the proposed method can still work well when the number of class increase, we conduct the experiments on CIFAR-100 and WebVision datasets. Specifically, for CIFAR-100, we keep the same experiment setting as CIFAR-10. For WebVision, we trained a standard ResNet-18 and an inception-resnet v2 (Szegedy et al., 2017) on WebVision for 1000 epochs using MoCo v2 to obtain the pretrained model. We follow the previous work (Chen et al., 2019) to select the first 50 classes of the Google image subset as the training set and leave out 10% of training examples as a noisy validation set. Then we train the model with the proposed CoNL. Other experiment settings

are as same as the experiment on CIFAR-10. We test the model on the human-annotated WebVision validation set. The test accuracy of CoNL with ResNet backbone is 64.88% and the test accuracy of CoNL with inception-resnet v2 backbone is 70.80%. The experiment results of CIFAR-100 are shown in Tab. 14.

Table 14: Test accuracy of CoNL on CIFAR-100.

|  | IDN-10% | IDN-20% | IDN-30% | IDN-40% | IDN-50% |
|---|---|---|---|---|---|
| CoNL-NR | $40.78 \pm 1.07$ | $39.94 \pm 1.51$ | $38.30 \pm 1.77$ | $36.25 \pm 1.69$ | $32.25 \pm 0.85$ |
| CoNL | $\mathbf{74.13 \pm 0.34}$ | $\mathbf{72.15 \pm 0.52}$ | $\mathbf{69.96 \pm 0.71}$ | $\mathbf{65.40 \pm 2.76}$ | $\mathbf{59.09 \pm 1.78}$ |

## C  DIFFERENCES BETWEEN CONTRASTIVE CO-SELECTING AND PREVIOUS WORK

Previous work Co-Teaching (Han et al., 2018) learn two classifiers to select confident examples for each other, and filter errors from the biased selection in the first mini-batch. In our work, two classifiers are only used in the co-selection stage and do not provide supervised signals for each other. We aim to design a method that can select examples as many as possible. Previous work AugDesc (Nishi et al., 2021) explores how to use data augmentation techniques for different permutations and combinations to improve the generalization and robustness of models without impacting the memorization effect negatively, *i.e.*, weak augmentation techniques for pseudo labels generation and strong augmentation techniques for the back-propagation step to update model's parameters. In this paper, different data augmentation strategies are used to enable contrastive co-selecting to select more reliable examples under different noise rates.

