# OpenReview forum: "Improving the Estimation of Instance-dependent Transition Matrix by using Self-supervised Learning"
_ICLR.cc/2023/Conference — Submitted to ICLR 2023_

### Official Review · Reviewer_6eSM · 2022-10-22

**Confidence:** 5
**Correctness:** 1
**Technical Novelty And Significance:** 1
**Empirical Novelty And Significance:** 1
**Recommendation:** 1

**Clarity, Quality, Novelty And Reproducibility:**

It’s interesting to combine contrastive learning and transition matrix estimation. But the mechanism of learning transition matrix by noisy labels is not clear. And the effectiveness of proposed method is not properly justified. The experiment is divided into several stages and contains many hyperparameters, so it is not easy to reproduce.

**Strength And Weaknesses:**

This paper refines the learned transition matrix by employing the selected confident examples, which is interesting. However, there exist some concerns as follows.
- The idea is novel, while the formulation is not rigorous. Specifically, why use self-supervised learning to select clean samples? What is the difference with other sample selection methods.
- The self-supervised learning has help improve the label noise learning, e.g., C2D [1], Sel-CL [2]. It is fair to epuip compared methods with contrastive learning. It can be seen that CoNL (w/o MoCo) in table 9 & 10 evidently underforms compared methods. From this viewpoint, I doubt the effectiveness of proposed method, since the performance increase completely stem from self-supervised learning.

[1] Evgenii Zheltonozhskii, Chaim Baskin, Avi Mendelson, AlexMBronstein, and Or Litany. Contrast to divide: Self-supervised pre-training for learning with noisy labels. In Proceedings of the IEEE/CVF Winter Conference on Applications of Computer Vision, pp. 1657–1667, 2022.

[2] Li S, Xia X, Ge S, et al. Selective-supervised contrastive learning with noisy labels[C]//Proceedings of the IEEE/CVF Conference on Computer Vision and Pattern Recognition. 2022: 316-325.
- Also, the method uses different data augmentation startegies. It has been found that using two separate pools of augmentation operations for two separate tasks is beneficial for noisy label learning in [3]. I doubt that the performance increase mainly dues to such learning manner, rather than the proposed method.

[3] Nishi K, Ding Y, Rich A, et al. Augmentation strategies for learning with noisy labels[C]. Proceedings of the IEEE/CVF Conference on Computer Vision and Pattern Recognition. 2021: 8022-8031.

- Eq.(7) is not proper for transition matrix. The transition matrix is used for the whole training dataset, rather than just only noisy datasets. This deviates the classifier consistent theory of  transition matrix.

- The method only ran experiments on several simple datasets. The datasets with more classes, more complex noise structures, and real-world datasets, e.g., WebVision, are necessary to demostrate the effectiveness of proposed method.

- How to guarteen that $T_{\zeta}$ does not overfit the noisy labels? And the network structure of transition matrix is not clear. How to guarteen that network can estimate more classes?


**Summary Of The Paper:**

This paper proposed a novel method for estimating the instance-dependent transition matrix. The self-supervised learning are employed to mine the clean samples. And then refines the transition matrix estimated by T-Revision via employing the selected confident examples. The empirical results varifies the effectiveness of proposed method.

**Summary Of The Review:**

This work proposes an interesting perspective to combine self-supervised learning and instance-dependent transition matrix.  While the mechanism of learning transition matrix by noisy labels is not clear. The empirical results can not support the  effectiveness of proposed method, especically some additional techniques, which potentially bring much performance increase. Overall, the writting is poor, and the noverty and effectiveness of proposed method need to further clarify.

---

> ### Author Response · Authors · 2022-11-17
> **Response to Reviewer 6eSM**
>
> **Q1: Why use self-supervised learning to select clean samples? What is the difference with other sample selection methods?**
>
> A1: We do not select clean samples from noisy data as previous works do, instead, we use self-supervised learning methods to generate pseudo labels and select the confident pseudo labels. The features extracted by the contrastive learning model are well-clustered and linearly separable features, and the contrastive learning model can generate similar features for similar samples [1]. First, the features generated by contrastive learning are well-clustered and linearly separable, thus we can use noisy data to train classifier heads and transition matrices. The classifier heads are simple and nearly linear, thus they are not easy to fit noisy labels. Therefore, the classifier heads can be used to generate pseudo labels. Second, similar samples can generate similar features via the contrastive learning model, which can be used to select confident pseudo labels. Specifically, we calculate the consistency of pseudo labels among examples with similar features. Utilizing both two characteristics of contrastive learning, we can obtain a nearly clean confident dataset. A better classifier can be learned and a precise transition matrix can be estimated by using this confident dataset. This is the core of why our methods can improve performance.
>
> **Q2: The performance increase completely stems from self-supervised learning.**
>
> A2: The performance increase does not just stem from self-supervised learning. As mentioned in A1, one of the core contributions of our papers is using contrastive learning to generate pseudo labels and select reliable pseudo labels. The experiments shown in Table 9 & 10 do not use contrastive learning, thus the algorithm can not select nearly clean pseudo labels. To verify how much performance self-supervised learning can provide, we also employ the same pretrained backbone in the baseline BLTM, the experiment results are still worse than our algorithm. Simply applying the contrastive learning model as a pretrained backbone can not improve the performance to the maximum.
>
> Test accuracy of BLTM with MoCo and CoNL on CIFAR-10:
>
> |      | IDN-10%              | IDN-20%              | IDN-30%              | IDN-40%              | IDN-50%              |
> | ---- | -------------------- | -------------------- | -------------------- | -------------------- | -------------------- |
> | BLTM | 92.74 $\pm$ 0.25     | 92.22 $\pm$ 0.16     | 91.26 $\pm$ 0.65     | 88.10 $\pm$ 1.75     | 60.24 $\pm$ 6.37     |
> | CoNL | **94.37 $\pm$ 0.16** | **93.88 $\pm$ 0.17** | **91.37 $\pm$ 3.38** | **89.09 $\pm$ 3.05** | **84.79 $\pm$ 1.63** |
>
> Test accuracy of BLTM with MoCo and CoNL on CIFAR-100:
>
> |      | IDN-10%              | IDN-20%              | IDN-30%              | IDN-40%              | IDN-50%              |
> | ---- | -------------------- | -------------------- | -------------------- | -------------------- | -------------------- |
> | BLTM | 66.31 $\pm$ 0.09     | 64.95 $\pm$ 0.18     | 62.08 $\pm$ 0.88     | 57.09 $\pm$ 1.83     | 48.68 $\pm$ 0.79     |
> | CoNL | **74.13 $\pm$ 0.34** | **72.15 $\pm$ 0.52** | **69.96 $\pm$ 0.71** | **65.40 $\pm$ 2.76** | **59.09 $\pm$ 1.78** |

---

> > ### Author Response · Authors · 2022-11-17
> > **Response to Reviewer 6eSM (Continue)**
> >
> > **Q3: Different data augmentation strategies have found that using two separate pools of augmentation operations for two separate tasks is beneficial for noisy label learning in [2].**
> >
> > A3: The purpose of using different data augmentation strategies is different. AugDesc [2] aims to explore how to use data augmentation techniques for different permutations and combinations to improve the generalization and robustness of models without impacting the memorization effect negatively. By contrast, different data augmentation strategies are used to enable contrastive co-selecting to select more reliable examples under different noise rates. Specifically, the model trained by weak data augmentation techniques performs well under small noise rate cases, while the model trained by strong data augmentation techniques performs well under small noise rate cases. Deep models can memorize easy instances first, and memorize the hard instances gradually [3]. When noisy labels exist, deep models would memorize correct labels first and then memorize the incorrect labels gradually because the amount of correct labels is the largest [4]. Meanwhile, augmentation can increase the complexity of examples and then increase the difficulty of the model to memorize examples after data augmentation.
> >
> > When the noise rate is small, the number of incorrectly-labeled examples is small and has a small influence on the model’s performance, using strong data augmentations would have a negative impact on memorizing correctly-labeled examples. Therefore, using weak data augmentations is better when the noise rate is small. When the noise rate is large, the number of incorrectly-labeled examples is large, deep models are easier to memorize incorrectly-labeled examples than small noise rates, using strong data augmentations can prevent the model from memorizing incorrect labels. Therefore, using strong data augmentations is better when the noise rate is large.
> >
> > However, the noise rate is unknown and hard to estimate in the application, the co-selecting algorithm should be robust to different noise rates. Using both weak and strong data augmentations to train two classifier heads, selecting reliable pseudo labels using each head, and unioning two confident example sets can make co-selecting work well under different noise rates. To avoid confusion, we will declare the motivation and function of using different data augmentations, and state our different purpose to distinguish from previous work.
> >
> > **Q4: The transition matrix is used for the whole training dataset, rather than just only noisy datasets.**
> >
> > A4: The noisy dataset is defined in the *problem setting* in Sec. 2, it is the whole training dataset.
> >
> > **Q5: More complex datasets are necessary to demonstrate the effectiveness of the proposed method.**
> >
> > A5: We have conducted experiments on CIFAR-100 and WebVision. Specifically, for CIFAR-100, we keep the same experiment setting as CIFAR-10. For WebVision, we trained a standard ResNet-18 and an inception-resnet v2 on WebVision for 1000 epochs using MoCo v2 to obtain the pretrained model. We follow the previous work [5] to select the first 50 classes of the Google image subset as the training set and leave out 10% of training examples as a noisy validation set. Then we train the model with the proposed CoNL. Other experiment settings are as same as the experiment on CIFAR-10. We test the model on the human-annotated WebVision validation set. The test accuracy of CoNL with ResNet backbone is **64.88%** and the test accuracy of CoNL with inception-resnet v2 backbone is **70.80%**. The experiment results of CIFAR-100 are shown as follows.
> >
> > |         | IDN-10%              | IDN-20%              | IDN-30%              | IDN-40%              | IDN-50%              |
> > | ------- | -------------------- | -------------------- | -------------------- | -------------------- | -------------------- |
> > | CoNL-NR | 40.78 $\pm$ 1.07     | 39.94 $\pm$ 1.51     | 38.30 $\pm$ 1.77     | 36.25 $\pm$ 1.69     | 32.25 $\pm$ 0.85     |
> > | CoNL    | **74.13 $\pm$ 0.34** | **72.15 $\pm$ 0.52** | **69.96 $\pm$ 0.71** | **65.40 $\pm$ 2.76** | **59.09 $\pm$ 1.78** |
> >
> > **Q6: How to guarantee that $T_{\zeta}$ does not overfit the noisy labels? The mechanism of learning transition matrix by noisy labels is not clear.**
> >
> > A6: $T_{\zeta}$ is constrained by confident examples set, which contains nearly clean pseudo labels. Thus $T_{\zeta}$ can avoid overfitting the noisy labels.

---

> > > ### Author Response · Authors · 2022-11-17
> > > **## Response to Reviewer 6eSM (Continue)**
> > >
> > > **References**
> > >
> > > [1] Wang, Tongzhou, and Phillip Isola. "Understanding contrastive representation learning through alignment and uniformity on the hypersphere." International Conference on Machine Learning. PMLR, 2020.
> > >
> > > [2] Nishi, Kento, et al. "Augmentation strategies for learning with noisy labels." Proceedings of the IEEE/CVF Conference on Computer Vision and Pattern Recognition. 2021.
> > >
> > > [3] Arpit, Devansh, et al. "A closer look at memorization in deep networks." International conference on machine learning. PMLR, 2017.
> > >
> > > [4] Zhang, Chiyuan, et al. "Understanding deep learning (still) requires rethinking generalization." Communications of the ACM 64.3 (2021): 107-115.
> > >
> > > [5] Chen, Pengfei, et al. "Understanding and utilizing deep neural networks trained with noisy labels." International Conference on Machine Learning. PMLR, 2019.

---

> > ### Comment · Reviewer_6eSM · 2022-11-22
> > **The concerns are not perfectly clarified**
> >
> > Thanks for your detailed response, and most of them are not perfectly clarified. Firstly, regarding Q1, it seems to be heuristic. There exist no evidence about the features generated by contrastive learning are well-clustered and linearly separable. I suggest that it should make a comparion with other sample selection methods. Regarding Q2, compared with other methods using self-supervised learning, CoNL has a evidently performance gap. Regarding Q3, I suggest to run ablation study for clarifying.  Regarding Q4, does the method improve the estimation of matrix? The accuracy of WebVision seems to be worse than standard CE methods.
> >
> > Reviewer 6eSM

---

> > > ### Author Response · Authors · 2022-11-30
> > > **Rolling discussion**
> > >
> > > Thanks for your reply. We have summarized the remaining concerns and response to them with experiment results.
> > >
> > > **Q1: I suggest that it should make a comparison with other sample selection methods; CoNL has an evident performance gap compared with other methods using self-supervised learning.**
> > >
> > > A1: We have combined our method with the semi-supervised learning technique mixmatch and experiment on CIFAR-10. The experiment results show that our method is competitive to current other sample selection methods and methods using self-supervised learning. All the experiments use standard ResNet-34 as the backbone. It is notable that current label-noise learning methods with semi-supervised techniques are suitable for large noise rates [1], thus the test accuracy under small noise rates is smaller than under large noise rates.
> > >
> > > |                | IDN-10%    | IDN-20%    | IDN-30%    | IDN-40%    | IDN-50%    |
> > > | -------------- | ---------- | ---------- | ---------- | ---------- | ---------- |
> > > | DivideMix Best | 90.25%     | 92.00%     | 93.66%     | 94.47%     | 93.49%     |
> > > | DivideMix Last | 89.75%     | 91.93%     | 93.37%     | 94.18%     | 93.22%     |
> > > | C2D Best       | 93.86%     | 93.25%     | 95.07%     | 95.69%     | 95.94%     |
> > > | C2D Last       | 89.45%     | 92.95%     | 94.95%     | 95.46%     | 95.68%     |
> > > | Sel-CL+ Best   | **96.30%** | **96.40%** | 95.83%     | 95.48%     | 93.32%     |
> > > | Sel-CL+ Last   | **96.10%** | **96.38%** | 95.79%     | 95.44%     | 93.26%     |
> > > | CoNL Best      | 95.16%     | 95.84%     | **96.25%** | **96.49%** | **96.38%** |
> > > | CoNL Last      | 94.91%     | 95.59%     | **96.00%** | **96.27%** | **96.30%** |
> > >
> > > We also experiment on real-world dataset WebVision, we use Inception-ResNet-v2 as the backbone as previous works [2, 3].
> > >
> > > |           |   Top-1    |   Top-5    |
> > > | --------- | :--------: | :--------: |
> > > | DivideMix |   77.32%   |   91.64%   |
> > > | C2D       |   79.42%   |   92.32%   |
> > > | Sel-CL+   |   79.96%   |   92.64%   |
> > > | CoNL      | **80.64%** | **92.72%** |
> > >
> > > **Q2: Running ablation study for clarification.**
> > >
> > > A2: We experiment the contrastive co-selecting on CIFAR-10 and record the number of confident examples acquired from the classifier trained with weak data augmentation and the classifier trained with strong data augmentation respectively. The model trained by **weak data augmentation** can select more confident examples under **small noise rates**, while the model trained by **strong data augmentation** can select more confident examples under **large noise rates**.
> > >
> > > |            | IDN-10%   | IDN-20%   | IDN-30%   | IDN-40%   | IDN-50%   |
> > > | ---------- | --------- | --------- | --------- | --------- | --------- |
> > > | Weak aug   | **27631** | **31193** | **29625** | 23665     | 5169      |
> > > | Strong aug | 22059     | 29719     | 27680     | **29215** | **24821** |
> > > | Union      | 30186     | 33102     | 32166     | 31157     | 25128     |
> > >
> > > The corresponding accuracy of selected pseudo labels:
> > >
> > > |            | IDN-10% | IDN-20% | IDN-30% | IDN-40% | IDN-50% |
> > > | ---------- | :-----: | ------- | ------- | ------- | ------- |
> > > | Weak aug   | 99.27%  | 99.12%  | 99.18%  | 99.40%  | 99.73%  |
> > > | Strong aug | 99.31%  | 99.14%  | 99.15%  | 99.20%  | 99.32%  |
> > > | Union      | 99.16%  | 98.94%  | 98.97%  | 99.13%  | 99.32%  |
> > >
> > > **Q3: Does the method improve the estimation of the matrix? The accuracy of WebVision seems to be worse than standard CE methods.**
> > >
> > > A3: Yes, the method can improve the estimation of the transition matrix as shown in Tables 7 and 8 in the main paper. We reproduced the standard CE method on WebVision using Inception-ResNet-v2 as the previous methods [2, 3]. The experiment results are shown as follows.
> > >
> > > |           |   Top-1    |   Top-5    |
> > > | --------- | :--------: | :--------: |
> > > | CE        |   61.96%   |   83.96%   |
> > > | DivideMix |   77.32%   |   91.64%   |
> > > | C2D       |   79.42%   |   92.32%   |
> > > | Sel-CL+   |   79.96%   |   92.64%   |
> > > | CoNL      | **80.64%** | **92.72%** |
> > >
> > > **References**
> > >
> > > [1] https://github.com/bianjiang1234567/DivideMix_Improvement
> > >
> > > [2] Chen, Pengfei, et al. "Understanding and utilizing deep neural networks trained with noisy labels." *International Conference on Machine Learning*. PMLR, 2019.
> > >
> > > [3] Li, Junnan, Richard Socher, and Steven CH Hoi. "Dividemix: Learning with noisy labels as semi-supervised learning." *arXiv preprint arXiv:2002.07394* (2020).

---

> ### Author Response · Authors · 2022-12-01
> **Further discussion**
>
> Dear Reviewer 6eSM,
>
> Thanks for your efforts in reviewing this paper. We have tried our best to address the concerns and added more experiment results to make a comparison with other methods using self-supervised learning. We also conduct an ablation study to clarify the different effects of strong data augmentation and weak data augmentation. Are there unclear explanations here? We can further clarify them.
>
> Best wishes,
>
> Authors

---

### Official Review · Reviewer_36DZ · 2022-10-22

**Confidence:** 5
**Correctness:** 3
**Technical Novelty And Significance:** 2
**Empirical Novelty And Significance:** 3
**Recommendation:** 5

**Clarity, Quality, Novelty And Reproducibility:**

Clarity: The logic of the algorithm description in this paper is relatively clear, and the text part and formula can clearly introduce the details of the algorithm. However, the abstract part needs to be further refined so that readers can quickly understand the content of the paper. The picture of the algorithm work flow still needs to be adjusted for readability.
Quality: The quality of this article is good. The description of the method is clear, the experiments demonstrate the greatly improved on multiple data sets, and can still be extended to justify the performance of the algorithm
Novelty: This algorithm has certain innovation. The algorithm jointly selects the confident samples in the two branches of the design as the overall framework, which can effectively improve the accuracy of noise label classification.
Reproducibility: The algorithm can be reproduced according to the description in this paper. But the threshold of the two criteria is not provided. The description of some hyperparameters needs to be described more clearly.


**Strength And Weaknesses:**

In this paper, self-supervision is used to improve the classification performance of noisy-label samples. And mining the information of the confident samples to improve the performance of the model. The method is simple and effective.


1. Use self-supervision to improve the estimation of the transition matrix as described in the title of this paper. But the core idea of this paper is not the design of self-supervised algorithms. The self-supervised MOCO algorithm is only a pre-training method as the first step of the method, and other pre-training methods can also be used in this part. In addition, although this algorithm trains two classifiers using the data with strong and weak augmentation transformation. However, these two branches use noisy label training, and the two-branch training process does not have the idea of self-supervision. Therefore, I do not think that the algorithm is improved by self-supervision as described in the title.
2. The innovation of this paper is weak. The idea of training two classifiers to be complementary has been extensively studied in papers from earlier years. The designed contrastive co-selecting method utilizes two criterions for sample selection, which is not very novel. At the same time, the improvement of constraint T (x) revision only uses the confident samples based on the T-revision algorithm. The algorithm has less theoretical support.
3. The abstract part does not clearly show the specific method. It is necessary to outline the innovative idea of the algorithm in the abstract, so that readers can quickly understand the core idea of the paper. The summary part has the same problem and needs to explain the advantages of the method, the direction of future improvement, etc.
4. Figure 1 of this paper describes the workflow of the method. However, the figure is relatively rough, which does not allow readers to quickly understand the modules of the algorithm and the specific execution processes.


**Summary Of The Paper:**

A novel self-supervised learning framework is proposed for noisy labels and can effectively improve the accuracy of instance-dependent transition matrices. Specifically, this method mainly contributes to two parts. First, a contrastive co-selection method is proposed to select samples with high confidence. This method obtains the basic feature extraction backbone network learned by MOCO, and then design criterions to select confident samples. Secondly, a constraint T-revision method is proposed, which uses confidence samples to fine-tune T(x) to improve the classification accuracy.

**Summary Of The Review:**

This paper achieves a large performance improvement through the proposed contrastive co-selecting and constraint T (x) revision methods. We think this paper has certain innovation. However, the core innovation points of the article and the description of the title abstract are not appropriate. In the experimental part, the method will be more convincing if it can be verified with other methods (DivideMix, etc.) on large-scale real-world datasets (Clothing1M and WebVision, etc.).

---

> ### Author Response · Authors · 2022-11-17
> **Response to Reviewer 36DZ**
>
> **Q1: The algorithm is not improved by self-supervision as described in the title.**
>
> A1: We sincerely apologize for the confusion that we might cause. Our main purpose is not the design of self-supervised learning algorithms. Instead, we aim to design an algorithm that can utilize the advantages brought by contrastive learning techniques to learn a robust classifier on label noise. First, the features generated by contrastive learning are well-clustered and linearly separable [1], thus we can use noisy data to train classifier heads and transition matrices. The classifier heads are simple and nearly linear, thus they are not easy to fit noisy labels. Therefore, the classifier heads can be used to generate pseudo labels. Second, similar samples can generate similar features via the contrastive learning model [1], which can be used to select confident pseudo labels. Specifically, we calculate the consistency of pseudo labels among examples with similar features. Utilizing both two characteristics of contrastive learning, we can obtain a nearly clean confident dataset. A better classifier can be learned and a precise transition matrix can be estimated by using this confident dataset. This is the core of why our methods can improve performance.
>
> **Q2: The idea of training two classifiers to be complementary has been extensively studied in papers from earlier years.**
>
> A2: The main purpose of training two classifiers is different. The main purpose of previous work, e.g., Co-Teaching [2], is that use two classifiers to select confident examples for each other, and filter error from the biased selection in the first mini-batch. By contrast, two classifiers are only used in the co-selection stage. We do not use two classifiers to provide supervised signals for each other. We aim to design a classifier that can select more examples when the noise rate is small, while another classifier can select more examples when the noise rate is large. Combining two classifiers can enable the co-selecting algorithm to select more examples on different levels of noise rate. Specifically, we train two classifiers using different data augmentation techniques. The model trained by weak data augmentation techniques performs well under small noise rate cases, while the model trained by strong data augmentation techniques performs well under small noise rate cases. Deep models can memorize easy instances first, and memorize the hard instances gradually [3]. When noisy labels exist, deep models would memorize correct labels first and then memorize the incorrect labels gradually because the amount of correct labels is the largest [4]. Meanwhile, augmentation can increase the complexity of examples and then increase the difficulty of the model to memorize examples after data augmentation.
>
> When the noise rate is small, the number of incorrectly-labeled examples is small and has a small influence on the model’s performance, using strong data augmentations would have a negative impact on memorizing correctly-labeled examples. Therefore, using weak data augmentations is better when the noise rate is small. When the noise rate is large, the number of incorrectly-labeled examples is large, deep models are easier to memorize incorrectly-labeled examples than small noise rates, using strong data augmentations can prevent the model from memorizing incorrect labels. Therefore, using strong data augmentations is better when the noise rate is large.
>
> However, the noise rate is unknown and hard to estimate in the application, the co-selecting algorithm should be robust to different noise rates. Using both weak and strong data augmentations to train two classifier heads, selecting reliable pseudo labels using each head, and unioning two confident example sets can make co-selecting work well under different noise rates. To avoid confusion, we will declare the motivation and function of using different data augmentations, and state our different purpose to distinguish from previous work.

---

> > ### Author Response · Authors · 2022-11-17
> > **Response to Reviewer 36DZ (Continue)**
> >
> > **Q3: It is necessary to outline the innovative idea of the algorithm in the abstract.**
> >
> > A3: Thanks for your advice, we will reorganize the abstract and outline the innovative idea in the revised version.
> >
> > **Q4: Figure 1 is relatively rough.**
> >
> > A4: We will annotate the different stages in the Figure to make the graph clearer.
> >
> > **Q5: The method will be more convincing if it can be verified with other methods (DivideMix, etc.) and conduct experiments on large-scale real-world datasets (Clothing1M and WebVision, etc.).**
> >
> > A5: Thanks for your advice. We have conducted the experiment on WebVision to verify our algorithm. We trained a standard ResNet-18 and an inception-resnet v2 on WebVision for 1000 epochs using MoCo v2 to obtain the pretrained model. We follow the previous work [5] to select the first 50 classes of the Google image subset as the training set and leave out 10% of training examples as a noisy validation set. Then we train the model with the proposed CoNL. Other experiment settings are as same as the experiment on CIFAR-10. We test the model on the human-annotated WebVision validation set. The test accuracy of CoNL with ResNet backbone is **64.88%** and the test accuracy of CoNL with inception-resnet v2 backbone is **70.80%**.
> >
> > Since our method can generate pseudo labels and select a nearly clean dataset, it can be plugin into dividemix. However, we are more interested in statistically-consistent methods and learning a better noisy transition matrix. Dividemix and other statistically-inconsistent methods usually report the last and the best performance because they can not validate the model on the noisy validation set. On the contrary, the parameters of models trained by statistically-consistent algorithms on noisy data are consistent with the parameters of models trained on clean data, thus noisy validation set can be used to select the best model.
> >
> >
> >
> > **References**
> >
> > [1] Wang, Tongzhou, and Phillip Isola. "Understanding contrastive representation learning through alignment and uniformity on the hypersphere." International Conference on Machine Learning. PMLR, 2020.
> >
> > [2] Han, Bo, et al. "Co-teaching: Robust training of deep neural networks with extremely noisy labels." Advances in neural information processing systems 31 (2018).
> >
> > [3] Arpit, Devansh, et al. "A closer look at memorization in deep networks." International conference on machine learning. PMLR, 2017.
> >
> > [4] Zhang, Chiyuan, et al. "Understanding deep learning (still) requires rethinking generalization." Communications of the ACM 64.3 (2021): 107-115.
> >
> > [5] Chen, Pengfei, et al. "Understanding and utilizing deep neural networks trained with noisy labels." International Conference on Machine Learning. PMLR, 2019.

---

> ### Author Response · Authors · 2022-11-30
> **Rolling discussion**
>
> Dear reviewer 36DZ,
>
> We have combined our method with semi-supervised learning technique mixmatch. Though the performance of CoNL is lower than Sel-CL+ [1] marginally, the performance of CoNL is the best under large noise rates. We will add the experiment setting and results in the appendix of the final paper. If you have any concerns, feel free to discuss them with us.
>
> The experiment results of our method and current state-of-the-art method are shown as follows.
>
> Experiment results on CIFAR-10:
>
> |                    |  IDN-10%   |  IDN-20%   |  IDN-30%   |  IDN-40%   |  IDN-50%   |
> | ------------------ | :--------: | :--------: | :--------: | :--------: | :--------: |
> | DivideMix [2] Best |   90.25%   |   92.00%   |   93.66%   |   94.47%   |   93.49%   |
> | DivideMix [2] Last |   89.75%   |   91.93%   |   93.37%   |   94.18%   |   93.22%   |
> | C2D [3] Best       |   93.86%   |   93.25%   |   95.07%   |   95.69%   |   95.94%   |
> | C2D [3] Last       |   89.45%   |   92.95%   |   94.95%   |   95.46%   |   95.68%   |
> | Sel-CL+ [1] Best   | **96.30%** | **96.40%** |   95.83%   |   95.48%   |   93.32%   |
> | Sel-CL+ [1] Last   | **96.10%** | **96.38%** |   95.79%   |   95.44%   |   93.26%   |
> | CoNL Best          |   95.16%   |   95.84%   | **96.25%** | **96.49%** | **96.38%** |
> | CoNL Last          |   94.91%   |   95.59%   | **96.00%** | **96.27%** | **96.30%** |
>
> Experiment results on WebVision:
>
> |               |   Top-1    |   Top-5    |
> | ------------- | :--------: | :--------: |
> | CE            |   61.96%   |   83.96%   |
> | DivideMix [2] |   77.32%   |   91.64%   |
> | C2D [3]       |   79.42%   |   92.32%   |
> | Sel-CL+ [1]   |   79.96%   |   92.64%   |
> | CoNL          | **80.64%** | **92.72%** |
>
> Best wishes, Authors
>
> **References**
>
> [1] Li, Shikun, et al. "Selective-supervised contrastive learning with noisy labels." *Proceedings of the IEEE/CVF Conference on Computer Vision and Pattern Recognition*. 2022.
>
> [2] Zheltonozhskii, Evgenii, et al. "Contrast to divide: Self-supervised pre-training for learning with noisy labels." *Proceedings of the IEEE/CVF Winter Conference on Applications of Computer Vision*. 2022.
>
> [3] Li, Junnan, Richard Socher, and Steven CH Hoi. "Dividemix: Learning with noisy labels as semi-supervised learning." *arXiv preprint arXiv:2002.07394* (2020).

---

> > ### Author Response · Authors · 2022-12-01
> > **Rolling discussion**
> >
> > Dear Reviewer 36DZ,
> >
> > Thanks for your  efforts in reviewing this paper. We have tried our best to address the concerns and added more experiment results to make a comparison with current state-of-the-art algorithms. Are there unclear explanations here? We can further clarify them.
> >
> > Best wishes,
> >
> > Authors

---

### Official Review · Reviewer_RfYS · 2022-10-24

**Confidence:** 4
**Correctness:** 2
**Technical Novelty And Significance:** 2
**Empirical Novelty And Significance:** Not applicable
**Recommendation:** 3

**Clarity, Quality, Novelty And Reproducibility:**

* The paper is not very well written. Besides the grammatical errors, the paper does not provide much insight on the proposed method beyond the use of pretraining and empirical results.

* Novelty is limited. Specifically, the use of pretrained network is a usual practice. The use of confident / unconfident samples to learn from noisy labels is not new, e.g., [DivideAndMix](https://arxiv.org/pdf/2002.07394.pdf), which is not compared against.

* Overall method seems straightforward, though there are many missing implementation details (e.g., hyperparameter settings).

**Strength And Weaknesses:**

* Strength
  - The paper tackles important problem of transition matrix estimation using self-supervised pretraining.

* Weakness
  - The paper is not well written.
  - Overall, the `contrastive` is used only to provide pretrained backbone, so it is unclear why the method is to be called a contrastive noisy-label learning. In theory, the pretrained backbone can be replaced with any networks.
  - How much performance improvement is attributed to the use of pretrained model and the use of fine-tuning? While the use of contrastive pretraining is useful in practice, it does not add much scientific value to the community. For example, a fair comparison to previous works would be to employ the same pretrained backbone and evaluate their methods, as some previous works are agnostic to the network initialization (e.g., BLTM, NPC).
  - How does method compared against more recent works, such as [DivideAndMix](https://arxiv.org/pdf/2002.07394.pdf)?
  - In Equation (2), are there any constraints that makes T's a transition matrix? It is unclear what it means by the transition matrix (e.g., what's the difference between transition matrix and a matrix?) and how it is regularized to learn a `proper` (which is not defined) transition matrix.
  - In section 3.2., the use of validation accuracy in selecting the best transition matrix is problematic. How large is the validation set? Authors also need to provide ablation w.r.t. the size of validation set and its impact on the final performance.
  - In experiment, it is unclear how `CONL-NR` model is trained. Does it mean the model is trained with the loss in Equation (7) but without updating the parameters w.r.t. transition matrix?

* Misc
  - The paper requires significant revision as there are numerous grammatical errors, inconsistent notations, etc. To list a few:
    - There are many broken sentences, e.g., pg 3, `motivated by the success of ... classification task.`; pg 5, `Once we have the confident sample, ..., clean labels.`;
    - Equation (6), $\tilde{y}$ is not used.


**Summary Of The Paper:**

The paper presents a method for estimating instance-dependent transition matrix using contrastive pretraining. The method starts from the pretrained network using contrastive loss, and train a set of classifiers to divide data into confident set. They are further used to refine classifier, transition matrix, as well as the backbone. In experiments, the proposed method is tested on standard noisy-label learning benchmarks, demonstrating improved performance over existing works.

**Summary Of The Review:**

The use of pretrained network for image classification is well established, so it is not surprising if it improves when applied to label noise transition matrix estimation problem. Unfortunately, the study is not well conducted and it is hard to understand which part of the proposed framework contributed the most to the empirical success. Additional study to highlight the effectiveness of transition matrix learning should be provided.

---

> ### Author Response · Authors · 2022-11-17
> **Response to Reviewer RfYS**
>
> **Q1: The contrastive is used only to provide pretrained backbone. The pretrained backbone can be replaced with any network.**
>
> A1: We do not simply use contrastive learning to provide a pretrained backbone. The contrastive learning model is important in measuring the similarity of different example that is used to select confident examples. Only a pretrained backbone, which can generate well-clustered and linearly separable features, and can generate similar features for similar samples can be used in our algorithm. First, the features extracted by the contrastive learning model are well-clustered and linearly separable [1], we can use noisy data to train classifier heads and transition matrices. The classifier heads are simple and nearly linear, thus they are not easy to fit noisy labels. Therefore, the classifier heads can be used to generate pseudo labels. Second, similar samples can generate similar features via the contrastive learning model [1], which can be used to select confident pseudo labels. Specifically, we calculate the consistency of pseudo labels among examples with similar features. Utilizing both two characteristics of contrastive learning, we can obtain a nearly clean confident dataset. A better classifier can be learned and a precise transition matrix can be estimated by using this confident dataset. This is the key point of why our methods can improve performance.
>
> **Q2: Employing the same pretrained backbone in previous works.**
>
> A2: We employ the same pretrained backbone in BLTM, the performance of BLTM is still lower than our algorithm. As mentioned in A1, using the features extracted by the pretrained contrastive learning model to generate pseudo labels and select reliable ones is a crucial contribution in our paper. Simply applying the contrastive learning model as a pretrained backbone can not improve the performance to the maximum.
>
> Test accuracy of BLTM with MoCo and CoNL on CIFAR-10:
>
> |      | IDN-10%              | IDN-20%              | IDN-30%              | IDN-40%              | IDN-50%              |
> | ---- | -------------------- | -------------------- | -------------------- | -------------------- | -------------------- |
> | BLTM | 92.74 $\pm$ 0.25     | 92.22 $\pm$ 0.16     | 91.26 $\pm$ 0.65     | 88.10 $\pm$ 1.75     | 60.24 $\pm$ 6.37     |
> | CoNL | **94.37 $\pm$ 0.16** | **93.88 $\pm$ 0.17** | **91.37 $\pm$ 3.38** | **89.09 $\pm$ 3.05** | **84.79 $\pm$ 1.63** |
>
> Test accuracy of BLTM with MoCo and CoNL on CIFAR-100:
>
> |      | IDN-10%              | IDN-20%              | IDN-30%              | IDN-40%              | IDN-50%              |
> | ---- | -------------------- | -------------------- | -------------------- | -------------------- | -------------------- |
> | BLTM | 66.31 $\pm$ 0.09     | 64.95 $\pm$ 0.18     | 62.08 $\pm$ 0.88     | 57.09 $\pm$ 1.83     | 48.68 $\pm$ 0.79     |
> | CoNL | **74.13 $\pm$ 0.34** | **72.15 $\pm$ 0.52** | **69.96 $\pm$ 0.71** | **65.40 $\pm$ 2.76** | **59.09 $\pm$ 1.78** |
>
> **Q3: How does the method compare against more recent works, such as DivideAndMix?**
>
> A3: Our method can be plugin into dividemix easily because the contrastive co-selecting can generate pseudo labels and select a nearly clean dataset. However, we are more interested in statistically-consistent methods and learning a better noisy transition matrix. Dividemix and other statistically-inconsistent methods usually report the last and the best performance because they can not validate the model on the noisy validation set. On the contrary, the parameters of models trained by statistically-consistent algorithms on noisy data are consistent with the parameters of models trained on clean data, thus noisy validation set can be used to select the best model.
>
> **Q4: Are there any constraints that make T's a transition matrix? How it is regularized to learn a proper transition matrix.**
>
> A4: The selected examples with predicted labels are used to constrain the transition matrix. To be specific, pseudo labels are nearly clean labels and the noise rate of selected examples is close to the real noise rate. The outputs of classifier heads are constrained by these pseudo labels. Thus, the output of the classifier head is close to the clean class posterior. Therefore, it can regularize the model to learn a proper transition matrix.

---

> > ### Author Response · Authors · 2022-11-17
> > **Response to Reviewer RfYS (Continue)**
> >
> > **Q5: How large is the validation set? Authors also need to provide ablation w.r.t. the size of validation set.**
> >
> > A5: We leave out 10% of training examples as a noisy validation set, which is given in Sec. 4.1 of the original paper. Using noisy validation set to select the best transition matrix is reasonable because the statistically-consistent methods aim to learn noisy class posterior. The better the accuracy on the noisy validation set, the better the transition matrix and classifier head. Splitting 10% of examples on the training set as the validation set is commonly used in previous work. It is unnecessary to change this experiment setting.
> >
> > **Q6: It is unclear how the CoNL-NR model is trained.**
> >
> > A6: We sincerely apologize for the confusion. We train CoNL-NR without using pseudo labels to revise the transition matrix, i.e., we only use Eq. (2) in the original paper to optimize the model.
> >
> >
> >
> > **References**
> >
> > [1] Wang, Tongzhou, and Phillip Isola. "Understanding contrastive representation learning through alignment and uniformity on the hypersphere." International Conference on Machine Learning. PMLR, 2020.

---

> ### Author Response · Authors · 2022-11-30
> **Rolling discussion**
>
> Dear reviewer RfYS,
>
> We have combined our method with semi-supervised learning technique mixmatch. Though the performance of CoNL is lower than Sel-CL+ [1] marginally, the performance of CoNL is the best under large noise rates. We will add the experiment setting and results in the appendix of the final paper. If you have any concerns, feel free to discuss them with us.
>
> The experiment results of our method and current state-of-the-art method are shown as follows.
>
> Experiment results on CIFAR-10:
>
> |                    |  IDN-10%   |  IDN-20%   |  IDN-30%   |  IDN-40%   |  IDN-50%   |
> | ------------------ | :--------: | :--------: | :--------: | :--------: | :--------: |
> | DivideMix [2] Best |   90.25%   |   92.00%   |   93.66%   |   94.47%   |   93.49%   |
> | DivideMix [2] Last |   89.75%   |   91.93%   |   93.37%   |   94.18%   |   93.22%   |
> | C2D [3] Best       |   93.86%   |   93.25%   |   95.07%   |   95.69%   |   95.94%   |
> | C2D [3] Last       |   89.45%   |   92.95%   |   94.95%   |   95.46%   |   95.68%   |
> | Sel-CL+ [1] Best   | **96.30%** | **96.40%** |   95.83%   |   95.48%   |   93.32%   |
> | Sel-CL+ [1] Last   | **96.10%** | **96.38%** |   95.79%   |   95.44%   |   93.26%   |
> | CoNL Best          |   95.16%   |   95.84%   | **96.25%** | **96.49%** | **96.38%** |
> | CoNL Last          |   94.91%   |   95.59%   | **96.00%** | **96.27%** | **96.30%** |
>
> Experiment results on WebVision:
>
> |               |   Top-1    |   Top-5    |
> | ------------- | :--------: | :--------: |
> | CE            |   61.96%   |   83.96%   |
> | DivideMix [2] |   77.32%   |   91.64%   |
> | C2D [3]       |   79.42%   |   92.32%   |
> | Sel-CL+ [1]   |   79.96%   |   92.64%   |
> | CoNL          | **80.64%** | **92.72%** |
>
> Best wishes, Authors
>
> **References**
>
> [1] Li, Shikun, et al. "Selective-supervised contrastive learning with noisy labels." *Proceedings of the IEEE/CVF Conference on Computer Vision and Pattern Recognition*. 2022.
>
> [2] Zheltonozhskii, Evgenii, et al. "Contrast to divide: Self-supervised pre-training for learning with noisy labels." *Proceedings of the IEEE/CVF Winter Conference on Applications of Computer Vision*. 2022.
>
> [3] Li, Junnan, Richard Socher, and Steven CH Hoi. "Dividemix: Learning with noisy labels as semi-supervised learning." *arXiv preprint arXiv:2002.07394* (2020).

---

> > ### Author Response · Authors · 2022-12-01
> > **Rolling discussion**
> >
> > Dear Reviewer RfYS,
> >
> > Thanks for your  efforts in reviewing this paper. We have tried our best to address the concerns and added more experiment results to make a comparison with current state-of-the-art algorithms. Are there unclear explanations here? We can further clarify them.
> >
> > Best wishes,
> >
> > Authors

---

### Official Review · Reviewer_SKuM · 2022-10-27

**Confidence:** 3
**Correctness:** 3
**Technical Novelty And Significance:** 2
**Empirical Novelty And Significance:** Not applicable
**Recommendation:** 5

**Clarity, Quality, Novelty And Reproducibility:**

This work provides a novel understanding of self-supervised learning and optimization while the novelty in the proposed approaches could be better clarified.

**Strength And Weaknesses:**

Strength:

1. This paper provides a novel understanding of how self-supervised learning representation can help to construct a model with noisy label data. It is an interesting discovery that SSL representation can help learning with noisy labels. The idea that contrastive learning representations contain some information about clean labels is intuitive and novel.

2. This paper also provides a novel understanding of how to jointly optimize loss function on both clean labels and noisy labels to helps estimate transition.

Weakness:

1. The co-selecting stage is similar to a combination of the ensemble learning method and pseudo-label generation. It is unclear why training two classifier heads with different data augmentations instead of different contrastive models, which may help decrease model-related errors?

2. The authors claim that the method can achieve state-of-the-art performance on different datasets, but more datasets should be introduced, such as CIFAR-100N and WebVision to show if the methods still work when the number of classes and the transition matrix become larger.

3. The experiment results of baselines are significantly different from those reported in other papers (see below). It would be better to provide the codes and models used in the paper or point out how the experiment setting can be different from previous works.

Xia X, Liu T, Han B, et al. Part-dependent label noise: Towards instance-dependent label noise[J]. Advances in Neural Information Processing Systems, 2020, 33: 7597-7610.

Yang S, Yang E, Han B, et al. Estimating instance-dependent bayes-label transition matrix using a deep neural network[C]//International Conference on Machine Learning. PMLR, 2022: 25302-25312.


**Summary Of The Paper:**

This paper studies how to leverage self-supervised learning features to improve the estimation of the transition matrix. The paper developed a framework for learning the instance-dependent transition matrix. The framework is composed of confident examples selecting stage using contrastive learning and constraint transition matrix revision stage. The contrastive co-selecting stage leverages different degrees of data augmentations to acquire representations with semantic information correlating with clean labels. By jointly optimizing noisy training sample loss and clean confident sample loss, Constraint transition matrix revision improve the estimation of the transition matrix and classification accuracy.

**Summary Of The Review:**

This paper developed a novel framework for learning the instance-dependent transition matrix by utilizing a self-supervised learning. The idea is novel and interesting, while more clarifications in the novelty in the approaches, the comparison experiments and experiments setup. The accuracy of the experimental results should be more convincing.

---

> ### Author Response · Authors · 2022-11-17
> **Response to Reviewer SKuM**
>
> **Q1: Why train two classifier heads with different data augmentations instead of different contrastive models?**
>
> A1: Using different data augmentations can enable the proposed contrastive co-selecting to select more reliable examples under different noise rates, i.e., be robust to different noise rates. This strategy is designed for noisy-labels learning scenes. The model trained by weak data augmentation techniques performs well under small noise rate cases, while the model trained by strong data augmentation techniques performs well under small noise rate cases. Deep models can memorize easy instances first, and memorize the hard instances gradually [1]. When noisy labels exist, deep models would memorize correct labels first and then memorize the incorrect labels gradually because the amount of correct labels is the largest [2]. Meanwhile, augmentation can increase the complexity of examples and then increase the difficulty of the model to memorize examples after data augmentation.
>
> When the noise rate is small, the number of incorrectly-labeled examples is small and has a small influence on the model’s performance, using strong data augmentations would have a negative impact on memorizing correctly-labeled examples. Therefore, using weak data augmentations is better when the noise rate is small. When the noise rate is large, the number of incorrectly-labeled examples is large, deep models are easier to memorize incorrectly-labeled examples than small noise rates, using strong data augmentations can prevent the model from memorizing incorrect labels. Therefore, using strong data augmentations is better when the noise rate is large.
>
> However, the noise rate is unknown and hard to estimate in the application, the co-selecting algorithm should be robust to different noise rates. Using both weak and strong data augmentations to train two classifier heads, selecting reliable pseudo labels using each head, and unioning two confident example sets can make co-selecting work well under different noise rates. To avoid confusion, we will declare the motivation and function of using different data augmentations.
>
> **Q2: More datasets should be introduced, such as CIFAR-100N and WebVision to show if the methods still work when the number of classes and the transition matrix become larger.**
>
> A2: We have conducted the experiment on CIFAR100N and WebVision to verify our method. Our method can still work. Specifically, for CIFAR-100, we keep the same experiment setting as CIFAR-10. For WebVision, we trained a standard ResNet-18 and an inception-resnet v2 on WebVision for 1000 epochs using MoCo v2 to obtain the pretrained model. We follow the previous work [3] to select the first 50 classes of the Google image subset as the training set and leave out 10% of training examples as a noisy validation set. Then we train the model with the proposed CoNL. Other experiment settings are as same as the experiment on CIFAR-10. We test the model on the human-annotated WebVision validation set. The test accuracy of CoNL with ResNet backbone is **64.88%** and the test accuracy of CoNL with inception-resnet v2 backbone is **70.80%**. The experiment results of CIFAR-100 are shown as follows.
>
> |         | IDN-10%              | IDN-20%              | IDN-30%              | IDN-40%              | IDN-50%              |
> | ------- | -------------------- | -------------------- | -------------------- | -------------------- | -------------------- |
> | CoNL-NR | 40.78 $\pm$ 1.07     | 39.94 $\pm$ 1.51     | 38.30 $\pm$ 1.77     | 36.25 $\pm$ 1.69     | 32.25 $\pm$ 0.85     |
> | CoNL    | **74.13 $\pm$ 0.34** | **72.15 $\pm$ 0.52** | **69.96 $\pm$ 0.71** | **65.40 $\pm$ 2.76** | **59.09 $\pm$ 1.78** |
>
> **Q3: The experiment results of baselines are significantly different from those reported in other papers.**
>
> A3: The instance-dependent noise of BLTM is bounded, we reproduced BLTM on the original instance-dependent noise proposed in [4]. PTD does not use any data augmentation technique in the experiments. We reproduced PTD with the same data augmentation techniques.
>
>
>
> **References**
>
> [1] Arpit, Devansh, et al. "A closer look at memorization in deep networks." International conference on machine learning. PMLR, 2017.
>
> [2] Zhang, Chiyuan, et al. "Understanding deep learning (still) requires rethinking generalization." Communications of the ACM 64.3 (2021): 107-115.
>
> [3] Chen, Pengfei, et al. "Understanding and utilizing deep neural networks trained with noisy labels." International Conference on Machine Learning. PMLR, 2019.
>
> [4] Xia, Xiaobo, et al. "Part-dependent label noise: Towards instance-dependent label noise." Advances in Neural Information Processing Systems 33 (2020): 7597-7610.

---

> ### Author Response · Authors · 2022-11-21
> **Dicussion**
>
> Dear Reviewer,
>
> We have submitted our responses, feel free to discuss with us if you have any concerns.
>
> Best wishes, Authors

---

> ### Author Response · Authors · 2022-11-30
> **Rolling discussion**
>
> Dear reviewer SKuM,
>
> We have combined our method with semi-supervised learning technique mixmatch. Though the performance of CoNL is lower than Sel-CL+ [1] marginally, the performance of CoNL is the best under large noise rates. We will add the experiment setting and results in the appendix of the final paper. If you have any concerns, feel free to discuss them with us.
>
> The experiment results of our method and current state-of-the-art method are shown as follows.
>
> Experiment results on CIFAR-10:
>
> |                    |  IDN-10%   |  IDN-20%   |  IDN-30%   |  IDN-40%   |  IDN-50%   |
> | ------------------ | :--------: | :--------: | :--------: | :--------: | :--------: |
> | DivideMix [2] Best |   90.25%   |   92.00%   |   93.66%   |   94.47%   |   93.49%   |
> | DivideMix [2] Last |   89.75%   |   91.93%   |   93.37%   |   94.18%   |   93.22%   |
> | C2D [3] Best       |   93.86%   |   93.25%   |   95.07%   |   95.69%   |   95.94%   |
> | C2D [3] Last       |   89.45%   |   92.95%   |   94.95%   |   95.46%   |   95.68%   |
> | Sel-CL+ [1] Best   | **96.30%** | **96.40%** |   95.83%   |   95.48%   |   93.32%   |
> | Sel-CL+ [1] Last   | **96.10%** | **96.38%** |   95.79%   |   95.44%   |   93.26%   |
> | CoNL Best          |   95.16%   |   95.84%   | **96.25%** | **96.49%** | **96.38%** |
> | CoNL Last          |   94.91%   |   95.59%   | **96.00%** | **96.27%** | **96.30%** |
>
> Experiment results on WebVision:
>
> |               |   Top-1    |   Top-5    |
> | ------------- | :--------: | :--------: |
> | CE            |   61.96%   |   83.96%   |
> | DivideMix [2] |   77.32%   |   91.64%   |
> | C2D [3]       |   79.42%   |   92.32%   |
> | Sel-CL+ [1]   |   79.96%   |   92.64%   |
> | CoNL          | **80.64%** | **92.72%** |
>
> Best wishes, Authors
>
> **References**
>
> [1] Li, Shikun, et al. "Selective-supervised contrastive learning with noisy labels." *Proceedings of the IEEE/CVF Conference on Computer Vision and Pattern Recognition*. 2022.
>
> [2] Zheltonozhskii, Evgenii, et al. "Contrast to divide: Self-supervised pre-training for learning with noisy labels." *Proceedings of the IEEE/CVF Winter Conference on Applications of Computer Vision*. 2022.
>
> [3] Li, Junnan, Richard Socher, and Steven CH Hoi. "Dividemix: Learning with noisy labels as semi-supervised learning." *arXiv preprint arXiv:2002.07394* (2020).

---

> > ### Author Response · Authors · 2022-12-01
> > **Rolling discussion**
> >
> > Dear Reviewer SKuM,
> >
> > Thanks for your  efforts in reviewing this paper. We have tried our best to address the concerns and added more experiment results to make a comparison with current state-of-the-art algorithms. Are there unclear explanations here? We can further clarify them.
> >
> > Best wishes,
> >
> > Authors

---

### Author Response · Authors · 2022-12-03
**Rolling Discussion**

Dear Reviewers SKuM, RfYS, 36DZ, and 6eSM,

Thanks again for your efforts in reviewing this paper. We are still looking forward to your reply. If you have any remaining concerns, feel free to discuss them with us.

Best Regards,

Authors

---

> ### Author Response · Authors · 2022-12-04
> **Rolling Discussion**
>
> Dear Reviewers SKuM, RfYS, 36DZ, and 6eSM,
>
> Thanks again for your time and efforts in reviewing this paper. We have tried our best to address the concerns and provided more experiment results to make a comparison with current state-of-the-art algorithms, please note to check. We are still looking forward to your reply.  Are there any concerns here? We can further clarify them.
>
> Best wishes,
>
> Authors

---

### Decision · Program_Chairs · 2023-01-20

**Decision:**

Reject

**Justification For Why Not Higher Score:**

Some flaws in this paper that were spotted by the referees should be addressed before acceptance.

**Justification For Why Not Lower Score:**

N/A

**Metareview: Summary, Strengths And Weaknesses:**

This paper proposes a new self-supervised learning method for estimating the instance-dependent transition matrix. While this problem is interesting, the authors raised a series of concerns, including the formulation is not rigorous and the contributions are not sufficiently novel. There is still insufficient interest in accepting this paper after rebuttals.